# A Unified Theoretical Analysis of Private and Robust Offline Alignment: from RLHF to DPO

**Xingyu Zhou** [1]  **Yulian Wu** [2]  **Francesco Orabona** [2]

## Abstract

In this paper, we theoretically investigate the effects of noisy labels in offline alignment, with a focus on the interplay between privacy and robustness against adversarial corruption. Specifically, under linear modeling assumptions, we present a unified analysis covering both reinforcement learning from human feedback (RLHF) and direct preference optimization (DPO) under different privacy-corruption scenarios, such as Local differential privacy-then-Corruption (LTC), where human preference labels are privatized before being corrupted by an adversary, and Corruption-then-Local differential privacy (CTL), where labels are corrupted before privacy protection. Our analysis leverages a reduction framework that reduces the offline alignment problem under linear modeling assumptions to parameter estimation in logistic regression. This framework allows us to establish an interesting separation result between LTC and CTL, demonstrating that LTC presents a greater challenge than CTL in offline alignment, even under linear models. As important by-products, our findings also advance the state-of-the-art theoretical results in offline alignment under privacy-only or corruption-only scenarios.

## 1. Introduction

The alignment training process in language models that utilizes a human-labeled preference dataset has been instrumental in producing more helpful, harmless, and honest responses (Bai et al., 2022). Leveraging an offline preference dataset, two prominent paradigms have emerged. The first is the *indirect* approach, such as Reinforcement Learning from Human Feedback (RLHF) (Ziegler et al., 2019; Ouyang et al., 2022), which learns an intermediate reward model before optimizing the policy. The second is the *direct* approach, exemplified by Direct Preference Optimization (DPO) (Rafailov et al., 2023), which directly optimizes the policy via supervised learning on the preference dataset.

It is clear that the performance of both RLHF and DPO is significantly influenced by the quality of the preference labels in the dataset. However, in practice, these labels are often noisy due to various factors (Lambert et al., 2023). One potential noise source is corruption or misspecification during label generation or data collection, e.g., data poisoning attack (Casper et al., 2023). Additionally, privacy concerns in human preference (as illustrated in Feng et al. (2024)) may prompt individuals to provide noisy or privatized preferences rather than their true rankings.

From a theoretical perspective, understanding the impact of these noisy labels—resulting from both corruption and privacy—is essential for improving offline alignment. Recent studies have made some initial attempts to address this issue (Mandal et al., 2024; Chowdhury et al., 2024; 2023; Bukharin et al., 2024), but they face two fundamental limitations: (1) They often treat corruption and privacy separately and focus exclusively on either RLHF or DPO, while, in practice, noisy labels can stem from both factors simultaneously; (2) The theoretical guarantees provided by these studies are often suboptimal, even when privacy and corruption are separately considered. Motivated by these limitations and practical scenarios, we are particularly interested in the following question:

*Can we provide a unified analysis of the interplay between privacy and robustness in both RLHF and DPO?*

We provide an affirmative answer to the above question by presenting the following contributions:

**1. A Unified Theoretical Framework.** We present a unified theoretical framework for analyzing the interplay between privacy and robustness in offline alignment, covering both RLHF and DPO. Specifically, for privacy protection, we consider *Local Differential Privacy (LDP)* (Kasiviswanathan et al., 2011; Duchi et al., 2013) for preference labels, while for robustness, we consider the *strong adversary corruption* model (Diakonikolas & Kane, 2023), where an adaptively chosen fraction of labels can be corrupted. Our frame-

[1]Wayne State University, USA [2]King Abdullah University of Science and Technology, Saudi Arabia. Correspondence to: Xingyu Zhou <xingyu.zhou@wayne.edu>.

*Proceedings of the 42nd International Conference on Machine Learning*, Vancouver, Canada. PMLR 267, 2025. Copyright 2025 by the author(s).

work can *simultaneously* handle three privacy-corruption scenarios for both RLHF and DPO: Corruption-then-LDP (CTL), LDP-then-Corruption (LTC), and Corruption-LDP-Corruption (CLC), capturing different ways privacy and corruption may interact in practice.

**2. Reduction to Logistic Regression.** Our unified analytical framework leverages a reduction that transforms the offline alignment problem, under certain linear modeling assumptions, into parameter estimation in logistic regression. This reduction enables us to establish suboptimality bounds for both RLHF and DPO by focusing on parameter estimation in logistic regression under private and corrupted labels across different scenarios. Moreover, it highlights key differences between RLHF and DPO, providing insights into practical design considerations.

**3. Separation between CTL and LTC.** A key takeaway from our study of the interplay between privacy and robustness to corruption is that LTC is a more challenging setting than CTL, illustrating that the order in which privacy and corruption interact with each other significantly impacts the performance of offline alignment.

**4. New State-of-the-art Guarantees.** Our results, when reduced to privacy-only or corruption-only settings, set new state-of-the-art results on theoretical guarantees for RLHF and DPO. For instance, for DPO under "corrupted" labels, our result is the first one that achieves $\mathcal{O}(1/\sqrt{n})$ rate (where $n$ is the size of preference dataset), matching the standard rate without noise. Additionally, as a by-product of our reduction approach, we provide the first results on parameter estimation error in logistic regression under both private and corrupted labels, which may be of independent interest.

Finally, we remark that, as in many previous related works, e.g., Zhu et al. (2023); Chowdhury et al. (2023), we consider linear modeling assumptions for the sake of theoretical analysis. However, we believe that our results could serve as important benchmarks for more general function classes. In fact, we have also verified our separation result between CTL and LTC in the general case via experiments on GPT2-large, see Appendix D for a detailed discussion.

## 2. Related Work

In the main body, we only focus on the most related work on robust and private offline alignment, while relegating an additional discussion to Appendix A.

**Provably robust alignment under corruption.** Mandal et al. (2024) considers offline RLHF with corrupted preference datasets and establishes upper bounds on the suboptimality gap under various coverage assumptions of the offline dataset. As will be discussed in Section 6.1, their results are either suboptimal or lack rigor due to gaps in their

proof. For robust DPO, Chowdhury et al. (2024) considers a strictly weaker corruption model and derives a suboptimality bound of rate $\mathcal{O}(1/n^{1/4})$. In contrast, our general result, when reduced to the same corruption model, achieves a better rate of $\mathcal{O}(1/\sqrt{n})$. Bukharin et al. (2024) also considers a specific corruption model in the label generation process of RLHF but only provides the estimation error of the reward model, without a performance guarantee for the final policy.

**Provably Private Alignment.** The most related work in this aspect is Chowdhury et al. (2023), which mainly focuses on the reward model estimation in RLHF under various privacy constraints (i.e., local and central label differential privacy). Our intermediate result on estimation error (Section 5) recovers the one in Chowdhury et al. (2023) when the corruption parameter is set to zero. Moreover, compared to the *implicit* suboptimality bound in Chowdhury et al. (2023), we provide the first explicit bound in terms of the *relative condition number* (Agarwal et al., 2021), which parallels similar results in standard (robust) offline RL (Zhang et al., 2022), i.e., reward-based rather than preference-based.

## 3. Preliminaries

**Background on Offline Alignment.** The goal of offline alignment is to further tune the Supervised Fine-Tuning (SFT) model to match human preferences using an offline preference dataset. The preference dataset $\mathcal{D} = (s_i, a_i^0, a_i^1, y_i)_{i=1}^n$ consists of $n$ samples, each has one context/state $s_i$ (e.g., prompt), two actions $a_i^0, a_i^1$ (e.g., two answers from language models) and label/preference feedback $y_i \in \{0, 1\}$ indicating which one is preferred by humans. We assume $s_i$ to be sampled independently from a distribution $\rho$. A widely used approach for modeling $y_i$ is Bradley-Terry model (Bradley & Terry, 1952):

$$\mathbb{P}\left\{y_i = l | s_i, a_i^0, a_i^1\right\} = \frac{\exp(r^\star(s_i, a_i^l))}{\exp(r^\star(s_i, a_i^0)) + \exp(r^\star(s_i, a_i^1))}, \quad (1)$$

for $l \in \{0, 1\}$, where $r^\star(\cdot, \cdot)$ is a ground truth reward model.

Based on this preference dataset, offline alignment aims to learn a good policy $\widehat{\pi}$. In particular, the performance of the learned policy $\widehat{\pi}$ is evaluated by the suboptimality gap between $\widehat{\pi}$ and a comparator policy $\pi^\dagger$, defined as

$$\text{SubOpt}(\widehat{\pi}, \pi^\dagger) = J(\pi^\dagger) - J(\widehat{\pi}), \quad (2)$$

where $J(\pi) := \mathbb{E}_{s \sim \rho, a \sim \pi(\cdot|s)}\left[r^\star(s, a)\right]$ and $\pi^\dagger$ is not necessarily the optimal policy.

**RLHF and DPO.** As already mentioned, there are two major paradigms in alignment for finding $\widehat{\pi}$: *indirect* and *direct* approaches. The former, exemplified by RLHF (Ziegler et al., 2019), involves an intermediate reward model learning process from preference dataset $\mathcal{D}$ before the policy optimization. The latter, represented by DPO (Rafailov et al.,

2023), employs a direct policy optimization, i.e., using a supervised-learning loss function to optimize the policy *directly* over the preference dataset $\mathcal{D}$.

**Privacy Protection in Human Feedback.** The preference signal $y_i$ in $\mathcal{D}$ could reveal sensitive personal information (Feng et al., 2024; Chowdhury et al., 2023), hence requiring a rigorous privacy protection. To this end, we consider the local label Differential Privacy (DP) (Chaudhuri & Hsu, 2011; Ghazi et al., 2021), which means that the learner now only has access to a privatized label rather than the raw one. More specifically, we have the following definition.

**Definition 3.1** (Label DP in Local Model (Chowdhury et al., 2023))**.** Let $\varepsilon > 0$ and $\delta \in [0, 1]$. If each label is privatized by a local randomizer $\mathcal{R}$, which satisfies for any $y, y'$ and any subset $S$ in the range of $\mathcal{R}$ that

$$\mathbb{P}\{\mathcal{R}(y) \in S\} \le e^{\varepsilon} \cdot \mathbb{P}\{\mathcal{R}(y') \in S\} + \delta,$$

then we say $\mathcal{R}$ is an $(\varepsilon, \delta)$-label differentially private local randomizer, and this privatized dataset is called label-private preference dataset. The entire alignment process that operates with the privatized dataset is said to satisfy local label DP. When $\delta = 0$, we simply say it is a $\varepsilon$-local label DP.

*Remark* 3.2 (Randomized Response)**.** Given the binary data of the true label, we would like to maintain the binary data property after privatization. Thus, we will adopt the standard randomized response mechanism (Warner, 1965) as our local randomizer, which essentially injects *controllable* noise in labels by a random flipping. Here, by "controllable," we mean the noise injection method, and noise level is under our control based on the privacy parameter $\varepsilon$.

**Corruption in Human Feedback.** The human feedback $y_i$ can often be noisy and even be corrupted in the source or during the data collection process, which deviates from the assumed true generation process in (1). To this end, the final learned policy $\widehat{\pi}$ needs to be robust with respect to corruption in labels. We consider a corruption model similar to *strong corruption model* from robust statistics literature (Diakonikolas & Kane, 2023), which roughly says that an adversary can adaptively corrupt the labels of a fraction of samples, by inspecting the samples.

**Definition 3.3** (Label Corruption Model)**.** Let $\alpha \in [0, 1/2]$. We consider an $\alpha$-corruption model: an adversary can *inspect* the samples in a preference dataset of size $n$ and then assign any label value of 0 or 1 to at most $\alpha n$ samples.

**Interplay between Privacy and Robustness.** One key theme of this paper is to study the interplay between privacy and robustness in offline alignment. In particular, we are interested in the impact of the *order* between privacy protection and corruption in the labels on the suboptimality gap (cf. (2)), for both RLHF and DPO. To this end, we will mainly consider the following settings.

**Definition 3.4** (CTL and LTC)**.** Given a raw preference dataset $\mathcal{D} = (s_i, a_i^0, a_i^1, y_i)_{i=1}^n$, we consider the following settings that differ in the order of privacy protection (see Definition 3.1) and corruption (see Definition 3.3). In all cases, the final input dataset for the learning algorithm will be denoted by $\mathcal{D}_{\text{in}} = (s_i, a_i^0, a_i^1, z_i)_{i=1}^n$.

**Corruption-then-LDP (CTL)**: An adversary first corrupts the labels in $\mathcal{D}$ to $\bar{y}_i$. Then, each label $\bar{y}_i$ is privatized by a local randomizer.

**LDP-then-Corruption (LTC)**: Each label $y_i$ in $\mathcal{D}$ is first privatized by a local randomizer, resulting in the private label $\widetilde{y}_i$. Then, the preference dataset with private labels is further corrupted by an adversary.

*Remark* 3.5. As a last setting, one may also consider the setting where corruption happens both before and after privacy protection, which turns out to be a simple combination of the results for CTL and LTC, hence omitted in our results.

## 4. Reduction to Parameter Estimation

In this section, we will show that the key to establishing the suboptimality guarantees in both RLHF and DPO is a tight parameter estimation in logistic regression, under certain modeling assumptions. This allows us to focus on a single-parameter estimation problem under different settings (i.e., CTL and LTC) for both RLHF and DPO. More importantly, this unified perspective also enables us to easily see the connection and difference between RLHF and DPO.

**Logistic Regression.** Recall that given a feature vector $x_i \in \mathbb{R}^d$, under logistic regression, the label $y_i \in \{0, 1\}$ is generated according to the following probability:

$$\mathbb{P}\{y_i = 1 | x_i\} = \sigma\left(\langle \theta_{\text{true}}, x_i \rangle\right), \tag{3}$$

where $\sigma(z) = \frac{1}{1+e^{-z}}$ is the sigmoid function, $\theta_{\text{true}} \in \mathbb{R}^d$ is the unknown true parameter and $\langle \cdot, \cdot \rangle$ denotes the inner product of two vectors.

### 4.1. RLHF with a Linear Reward Model

We show that when the reward model in (1) is a linear function, the key to bounding the suboptimality gap in RLHF is the parameter estimation in a logistic regression problem. To start with, we formally state the linear reward model, following common definitions used in prior work (Zhu et al., 2023; Xiong et al., 2024; Cen et al., 2024; Chowdhury et al., 2023; Mandal et al., 2024).

**Assumption 4.1** (Linear Reward with Boundedness)**.** We assume that the ground truth reward $r^\star$ is linear, i.e., $r^\star(s, a) = \langle \phi(s, a), \theta^\star \rangle$, where $\phi(s, a) : \mathcal{S} \times \mathcal{A} \to \mathbb{R}^d$ is some known and fixed feature map and $\mathcal{S}$, $\mathcal{A}$ are the state space and the action space, respectively. We also assume the following standard boundedness conditions. For

all $s \in \mathcal{S}$ and $a \in \mathcal{A}$, without loss of generality, we assume $\|\phi(s,a)\| \leq 1$. Moreover, we assume $\theta^\star \in \Theta_B = \{\theta \in \mathbb{R}^d : \langle 1, \theta \rangle = 0, \|\theta\| \leq B\}$, where the condition $\langle 1, \theta \rangle = 0$ is to ensure the identifiability of $\theta^\star$.

Under the above assumption, we consider the standard offline RLHF algorithm, but with an additional parameter $\eta$. In particular, we consider two alternative outputs: When $\eta = 0$, the output policy is $\widehat{\pi} = \operatorname{argmax}_\pi \widehat{J}(\pi)$ where $\widehat{J}(\pi) = \mathbb{E}_{s \sim \rho, a \sim \pi(\cdot|s)}[\langle \widehat{\theta}, \phi(s,a) \rangle]$, that is essentially a greedy algorithm with respect to an estimate $\widehat{\theta}$; When $\eta = 1$, the output is $\widehat{\pi} = \operatorname{argmax}_\pi \widehat{J}(\pi)$, where the objective function is defined via the principle of pessimism (Zhu et al., 2023; Jin et al., 2021; Li et al., 2024) as

$$\widehat{J}(\pi) = \min_{\theta \in \Theta(\widehat{\theta}, \lambda)} \mathbb{E}_{s \sim \rho, a \sim \pi(\cdot|s)}[\langle \theta, \phi(s,a) \rangle]$$
$$- \mathbb{E}_{s \sim \rho, a \sim \pi_{\mathrm{ref}}(\cdot|s)}[\langle \theta, \phi(s,a) \rangle],$$

by constructing a confidence set around an estimate $\widehat{\theta}$:

$$\Theta(\widehat{\theta}, \lambda) = \left\{ \theta \in \Theta_B \mid \left\| \widehat{\theta} - \theta \right\|_{\widehat{\Sigma} + \lambda I} \leq \Gamma(n, d, \delta, \lambda) \right\} .$$

For completeness and due to space limitations, the full algorithm is given in Algorithm 2 in the Appendix B.

Here, we use a reference policy $\pi_{\mathrm{ref}}$ because the confidence set only measures the uncertainty of the difference in reward. That is, it does not measure the uncertainty for a single state-action pair.

We have the following key theoretical result on Algorithm 2, with its proof in Appendix E.1.

**Proposition 4.2.** *Under Assumption 4.1, the labels $\{y_i\}_{i \in [n]}$ in the preference dataset of RLHF follow the logistic regression model with $\theta_{\mathrm{true}} = \theta^\star$ and $x_i = \phi(s_i, a_i^1) - \phi(s_i, a_i^0)$. Algorithm 2 with $\eta = 0$ achieves*

$$\mathrm{SubOpt}(\widehat{\pi}, \pi^\star) \leq 2 \left\| \widehat{\theta} - \theta_{\mathrm{true}} \right\|_2 , \qquad (4)$$

*where $\pi^\star = \operatorname{argmax}_\pi J(\pi)$. Further, let $\widehat{\Sigma} := \frac{1}{n} \sum_i x_i x_i^\top$ and $\lambda > 0$ and suppose with probability at least $1 - \delta$ the estimate $\widehat{\theta}$ satisfies*

$$\left\| \widehat{\theta} - \theta_{\mathrm{true}} \right\|_{\widehat{\Sigma} + \lambda \mathbf{I}} \leq \Gamma(n, d, \delta, \lambda) . \qquad (5)$$

*Then, setting $\eta = 1$ in Algorithm 2, we have for any $\pi^\dagger$ and $\rho$, with probability at least $1 - \delta$,*

$$\mathrm{SubOpt}(\widehat{\pi}, \pi^\dagger) \leq 2\Gamma(n, d, \delta, \lambda)$$
$$\times \left\| \mathbb{E}_{s \sim \rho}[\phi(s, \pi^\dagger(s)) - \phi(s, \pi_{\mathrm{ref}}(s))] \right\|_{(\widehat{\Sigma} + \lambda I)^{-1}} , \quad (6)$$

*for any reference policy $\pi_{\mathrm{ref}}$, where we define $\phi(s, \pi(s)) := \mathbb{E}_{a \sim \pi(\cdot|s)}[\phi(s,a)]$.*

We can further simplify the result in (6) by introducing the following *relative condition number*, which can be viewed as the natural extension of standard one (Zhang et al., 2022; Agarwal et al., 2021) to the RLHF setting.

**Definition 4.3** (Relative Condition Number). For $\pi_1, \pi_2$ and a feature map $\phi$, we define $\psi(s, a, a') = \phi(s, a) - \phi(s, a')$ and $\Sigma_{\pi_1, \pi_2}$ as

$$\mathbb{E}_{s \sim \rho, a \sim \pi_1(\cdot|s), a' \sim \pi_2(\cdot|s)} \psi(s, a, a') \psi(s, a, a')^\top . \quad (7)$$

For any comparator policy $\pi^\dagger$ and any given reference policy $\pi_{\mathrm{ref}}$, we define

$$\kappa(\pi^\dagger, \pi_{\mathrm{ref}}) := \sup_{w \in \mathbb{R}^d} \frac{w^\top \Sigma_{\pi^\dagger, \pi_{\mathrm{ref}}}^{\mathrm{diff}} w}{w^\top \Sigma_{\pi_{\mathrm{sft}}, \pi_{\mathrm{sft}}}^{\mathrm{diff}} w} . \quad (8)$$

We can now simplify our previous suboptimality bound using the relative condition number above in the following corollary, with its proof given by Appendix E.2.

**Corollary 4.4.** *Let the same assumption in Proposition 4.2 hold and further assume $\lambda \geq \Omega\left(\frac{d}{n} \cdot \ln(n/\delta)\right)$. For any given comparator policy $\pi^\dagger$ with $\kappa(\pi^\dagger, \pi_{\mathrm{ref}}) < \infty$, we can upper bound (6) as follows:*

$$\mathrm{SubOpt}(\widehat{\pi}, \pi^\dagger) \leq 2\sqrt{3} \cdot \Gamma(n, d, \delta, \lambda) \cdot \sqrt{d \cdot \kappa(\pi^\dagger, \pi_{\mathrm{ref}})}.$$

### 4.2. DPO with a Log-Linear Policy Class

In this section, we will show that for a log-linear policy class (defined below), the suboptimality in DPO is also related to the parameter estimation in logistic regression.

We begin with a brief recap of DPO, following the original paper (Rafailov et al., 2023). The key idea is to reparameterize the reward model by the optimal policy of a KL-regularized problem. In particular, for the following KL-regularized optimization objective (with $\beta > 0$)

$$J_\beta(\pi) = \mathbb{E}_{s \sim \rho, a \sim \pi(\cdot|s)} \left[ r^\star(s, a) - \beta \ln \frac{\pi(a|s)}{\pi_{\mathrm{sft}}(a|s)} \right],$$

the optimal solution has the closed-form expression

$$\pi^\star(a|s) = \frac{1}{Z_\beta(s)} \pi_{\mathrm{sft}}(a|s) \exp(r^\star(s,a)/\beta), \quad (9)$$

where $Z_\beta(s) = \sum_{a \in \mathcal{A}} \pi_{\mathrm{sft}}(a|s) \exp(r^\star(s,a)/\beta)$ is the normalization factor. This allows us to rewrite the reward $r^\star$ in terms of $\pi^\star$ as follows

$$r^\star(s,a) = \beta \ln \frac{\pi^\star(a|s)}{\pi_{\mathrm{sft}}(a|s)} + \beta \ln Z_\beta(s) . \quad (10)$$

With the above re-parametrization of the reward using policy in (10) and BT preference model in (1), DPO (Rafailov et al.,

2023) directly minimizes the following log-loss function:

$$\mathcal{L}(\pi; \pi_{\mathrm{sft}}) :=$$

$$- \sum_{i=1}^{n} \mathbb{1}(y_i = 0) \ln \sigma \left( \beta \ln \frac{\pi(a_i^0|s_i)}{\pi_{\mathrm{sft}}(a_i^0|s_i)} - \beta \ln \frac{\pi(a_i^1|s_i)}{\pi_{\mathrm{sft}}(a_i^1|s_i)} \right)$$

$$- \sum_{i=1}^{n} \mathbb{1}(y_i = 1) \ln \sigma \left( \beta \ln \frac{\pi(a_i^1|s_i)}{\pi_{\mathrm{sft}}(a_i^1|s_i)} - \beta \ln \frac{\pi(a_i^0|s_i)}{\pi_{\mathrm{sft}}(a_i^0|s_i)} \right) .$$

$$(11)$$

In this paper, we consider the log-linear policy class for the sake of theoretical analysis.

**Assumption 4.5** (Log-linear Policy Class). We assume that the optimal policy in (9) satisfies $\pi^\star \in \Pi$ and $\pi_{\mathrm{sft}} \in \Pi$ where

$$\Pi = \left\{ \pi_\theta(a|s) = \frac{\exp(\langle \theta, \phi(s,a) \rangle)}{\sum_{a' \in \mathcal{A}} \exp(\langle \theta, \phi(s,a') \rangle)} \right\}, \quad (12)$$

is the log-linear class for some known feature map $\phi(s,a)$ : $\mathcal{S} \times \mathcal{A} \to \mathbb{R}^d$ with $\|\phi(s,a)\| \leq 1$. Moreover, $\theta^\star$ corresponding to $\pi^\star$ satisfies that $\theta^\star \in \Theta_B = \{\theta \in \mathbb{R}^d : \langle 1, \theta \rangle = 0, \|\theta\| \leq B\}$, where the condition $\langle 1, \theta \rangle = 0$ is to ensure the identifiability of $\theta^\star$.

The above policy realizability assumption is equivalent to the reward model realizability. In particular, by plugging log-linear policy into (11), we can establish that the labels $y_i$ again follow from the logistic regression in (3) with proper choices of $\theta_{\mathrm{true}}$ and $x_i$. In particular, we have the following formal statement, with its proof in Appendix E.3.

**Proposition 4.6.** *Under Assumption 4.5, the labels $\{y_i\}_{i \in [n]}$ in the preference dataset of DPO follow the logistic regression model with $\theta_{\mathrm{true}} = \beta(\theta^\star - \theta_{\mathrm{sft}})$ with $\beta > 0$ and $x_i = \phi(s_i, a_i^1) - \phi(s_i, a_i^0)$. Suppose with probability at least $1 - \delta$, there exists an estimate $\widehat{\theta}$ that satisfies*

$$\left\| \widehat{\theta} - \theta_{\mathrm{true}} \right\|_{\widehat{\Sigma} + \lambda \mathbf{I}} \leq \Gamma(n, d, \delta, \lambda), \quad (13)$$

*where $\widehat{\Sigma} := \frac{1}{n} \sum_i x_i x_i^\top$ and $\lambda > 0$. Then, let $\widehat{\theta}' = \widehat{\theta}/\beta + \theta_{\mathrm{sft}}$ and $\lambda \geq \Omega\left(\frac{d}{n} \cdot \ln(n/\delta)\right)$, the corresponding policy $\widehat{\pi} = \pi_{\widehat{\theta}'}$ with probability at least $1 - \delta$ satisfies*

$$\mathrm{SubOpt}(\widehat{\pi}, \pi^\star) \leq \frac{\sqrt{3}}{\sqrt{2}} \cdot \sqrt{\kappa_\Pi} \cdot B \cdot \Gamma(n, d, \delta, \lambda),$$

*where $\kappa_\Pi := \max_{\pi \in \Pi} \kappa(\pi, \pi)$ is the maximum relative condition number across the entire policy class.*

*Remark* 4.7. One can also rewrite the above bound using the maximum value of the implicit reward function, $r_{\max}$ as

$$\mathrm{SubOpt}(\widehat{\pi}, \pi^\star) \leq c \cdot \sqrt{\kappa_\Pi} \cdot \frac{r_{\max}}{\beta} \cdot \Gamma(n, d, \delta, \lambda),$$

for some constant $c > 0$ and log-linear policy $\Pi$.

*Remark* 4.8 (single-policy vs. all-policy concentrability). One nice thing about the above reduction is that it allows us to easily see the key difference between RLHF and DPO. In particular, from Corollary 4.4 and Proposition 4.6, we can see that the key (and only) difference lies in the choice of relative condition number (especially when considering the typical scaling of $B = \mathcal{O}(\sqrt{d})$ for the parameter), which is also closely related to the "concentratability coefficient" in offline RL (Munos, 2007; Jin et al., 2021). In particular, due to the use of pessimism in offline RLHF, one can achieve a bound in terms of $\kappa(\pi^\dagger, \pi_{\mathrm{ref}})$, which is related to the "single-policy concentratability" (Rashidinejad et al., 2021; Jin et al., 2021) for *any* comparator policy $\pi^\dagger$. On the other hand, due to the lack of uncertainty characterization in DPO, one needs "all-policy concentratability" (Chen & Jiang, 2019) $\kappa_\Pi$ in the upper bound, which is often much larger. In fact, this kind of dependence in standard DPO is shown to be necessary (Song et al., 2024).

# 5. Parameter Estimation Under Private and Corrupted Labels

As motivated by the last section, we now turn to designing algorithms for providing label privacy while accurately estimating the unknown parameter $\theta_{\mathrm{true}}$ in logistic regression, even under corrupted labels. As we will see, the key to the design is a new loss function, which allows us to adaptively handle the privacy-robustness interplays in a unified way. To facilitate the upcoming discussion, we formally state the general problem setup for logistic regression under private and corrupted labels.

**Definition 5.1** (Private and robust parameter estimation problem). Let $\mathcal{D}$ be a dataset of i.i.d samples $\{x_i, y_i\}_{i=1}^n$ where $x_i \sim \mu$ and $y_i$ follows from the logistic regression model in (3). The input dataset $\mathcal{D}_{\mathrm{in}} = \{x_i, z_i\}_{i=1}^n$ is the private and corrupted version of $\mathcal{D}$, following Definition 3.4. The goal here is to design a local randomizer $\mathcal{R}$ for privatizing labels (cf. Definition 3.1) as well as an analyzer $\mathcal{A}$ that receives $\mathcal{D}_{\mathrm{in}}$ outputs an estimate $\widehat{\theta}$ that is close to the underlying true parameter $\theta_{\mathrm{true}}$, measured by a proper choice of norm. We assume the following boundedness conditions: for any $i \in [n]$, $\|x_i\| \leq 1$ and $\theta_{\mathrm{true}} \in \Theta_{B'} = \{\theta \in \mathbb{R}^d : \langle 1, \theta \rangle = 0, \|\theta\| \leq B'\}$.

*Remark* 5.2. The boundedness assumption essentially follows from the reduction in the last section. Here, we assume $\|x_i\| \leq 1$ rather than upper bounded by 2 for simplicity and $B'$ can be properly chosen for RLHF and DPO, respectively.

## 5.1. Our Algorithm

As mentioned, our choice of local randomizer $\mathcal{R}$ for privacy protection is the simple Random Response (RR) mechanism with parameter $\varepsilon > 0$ (Warner, 1965). That is, the binary output from RR equals the input with probability $\sigma(\varepsilon) =$

---

**Algorithm 1** Private and Robust Estimation

---

1: **Procedure:** $\varepsilon$-local label DP mechanism $\mathcal{R}$
2: //Input: $\quad U_i \in \{0,1\}$, parameter: $\varepsilon$
3: Random response: $\widetilde{U}_i = \begin{cases} U_i & w.p. \ \frac{e^\varepsilon}{e^\varepsilon+1} \\ 1-U_i & w.p. \ \frac{1}{e^\varepsilon+1} \end{cases}$
4: **Return** $\widetilde{U}_i$
5: **Procedure:** Analyzer $\mathcal{A}$
6: //Input: $\{(x_i, z_i)\}_{i=1}^n$, parameter: $\varepsilon$
7: Let $c(\varepsilon) = \frac{1}{2\sigma(\varepsilon)-1} = \frac{e^\varepsilon+1}{e^\varepsilon-1}$
8: Compute $\widehat{\theta} = \arg\min_{\theta \in \Theta_{B'}(\theta)} -\frac{1}{n}\sum_{i=1}^n \widetilde{\ell}_i(\theta)$ where

$$\widetilde{\ell}_i(\theta) = \ln(1 - \sigma(\theta^\top x_i)) + (z_i + \sigma(\varepsilon) - 1)c(\varepsilon)\theta^\top x_i$$

9: **Return** $\widehat{\theta}$

---

$\frac{e^\varepsilon}{1+e^\varepsilon}$; otherwise, the privatized binary output differs from the input. RR satisfies the $\varepsilon$-local label DP guarantee (cf. Definition 3.1) (Dwork & Roth, 2014).

We now turn to the design of the analyzer $\mathcal{A}$, which is responsible for outputting an estimate $\widehat{\theta}$. We first point out that in the non-private non-corrupted case, the standard maximum likelihood estimator (MLE) that minimizes the loss function $\mathcal{L}(\theta) = -\frac{1}{n}\sum_{i=1}^n \ell_i(\theta)$ enjoys a good concentration (Zhu et al., 2023) with respect to $\theta_{\text{true}}$, where $\ell_i(\theta)$ is the standard log-loss:

$$\ell_i(\theta) = y_i \log(\sigma(\theta^\top x_i)) + (1 - y_i)\log(1 - \sigma(\theta^\top x_i))$$
$$= \log(1 - \sigma(\theta^\top x_i)) + y_i \theta^\top x_i.$$

However, due to the private labels, our analyzer is designed to minimize a new loss $\widetilde{\mathcal{L}}(\theta) = -\frac{1}{n}\sum_{i=1}^n \widetilde{\ell}_i(\theta)$ where

$$\widetilde{\ell}_i(\theta) = \ln(1 - \sigma(\theta^\top x_i)) + (z_i + \sigma(\varepsilon) - 1)c(\varepsilon)\theta^\top x_i, \quad (14)$$

and $c(\varepsilon) := \frac{1}{2\sigma(\varepsilon)-1} = \frac{e^\varepsilon+1}{e^\varepsilon-1}$. The key difference lies in the "shifting and scaling" of the received labels $z_i$, which, in fact, enjoys exactly the same "shifting and scaling" intuition as in mean estimation under RR, i.e., it is an unbiased estimate. Putting the above choices of $\mathcal{R}$ and $\mathcal{A}$ together, yields the final Algorithm 1 above.

*Remark* 5.3. We remark that a similar loss (up to some scaling) has been considered in Chowdhury et al. (2023; 2024). However, they are motivated from a different perspective (e.g., logit) rather than our connection to standard mean estimation under RR for local privacy (i.e., shifting and scaling). The form we use here in (14) has not appeared before. This new form not only makes it easy to see that our new loss is an unbiased estimate of the standard log loss, but also allows us to easily show that our single algorithm is *adaptive* to different privacy-corruption settings, i.e., it does not know the specific setting in advance.

## 5.2. Estimation Error Bounds

In this section, we will establish the estimation error bounds achieved by Algorithm 1. Throughout this section, we will let $\widehat{\theta}_{\text{CTL}}, \widehat{\theta}_{\text{LTC}}$ be the estimates outputted by Algorithm 1 under CTL and LTC respectively. Our first result is the following theorem, which characterizes the estimator error in terms of a weighted norm, with proof in Appendix E.4.

**Theorem 5.4.** *Consider the problem in Definition 5.1. For any $\varepsilon > 0$, $\alpha, \in [0, 1/2]$, $\delta \in (0,1)$, and $\lambda > 0$, with probability at least $1-\delta$, the output of Algorithm 1 achieves*

$$\left\|\widehat{\theta}_{\text{CTL}} - \theta_{\text{true}}\right\|_{\widehat{\Sigma}+\lambda\mathbf{I}} \leq \Gamma_{\text{CTL}}(n, d, \delta, \lambda)$$
$$:= C\left(\frac{\sqrt{\alpha}}{\gamma} + \frac{c(\varepsilon)}{\gamma}\sqrt{\frac{d + \ln(1/\delta)}{n}} + B'\sqrt{\lambda}\right),$$

$$\left\|\widehat{\theta}_{\text{LTC}} - \theta_{\text{true}}\right\|_{\widehat{\Sigma}+\lambda\mathbf{I}} \leq \Gamma_{\text{LTC}}(n, d, \delta, \lambda)$$
$$:= C\left(\frac{c(\varepsilon)\sqrt{\alpha}}{\gamma} + \frac{c(\varepsilon)}{\gamma}\sqrt{\frac{d + \ln(1/\delta)}{n}} + B'\sqrt{\lambda}\right),$$

*where $\widehat{\Sigma} = \frac{1}{n}\sum_{i=1}^n x_i x_i^\top$, $c(\varepsilon) = \frac{e^\varepsilon+1}{e^\varepsilon-1}$, $\gamma = 1/(2 + \exp(-B') + \exp(B'))$, and $C$ is a universal constant.*

*Remark* 5.5. First, when there is no corruption, our result matches the one in previous work on private parameter estimation (Chowdhury et al., 2023). Second, *when corruption exists, the order of corruption and local privacy matters*. In particular, LTC has an additional cost $c(\varepsilon)$ in the first corruption term compared to CTL, highlighting the interplay between privacy and robustness.

Our second result is a concentration result under $L_2$-norm with the additional condition of *uniform coverage*, which has been leveraged in prior work as well (Mandal et al., 2024; Zhang et al., 2022; Chowdhury et al., 2023).

**Assumption 5.6** (Uniform Coverage). There exists a positive constant $\xi > 0$ such that the minimum eigenvalue $\lambda_{\min}(\Sigma) \geq \xi$, where $\Sigma := \mathbb{E}_{x\sim\mu}[xx^\top]$.

Under the above assumption, we can have another estimation error bound for the underlying parameter, which is now in terms of $L_2$-norm, with proof in Appendix E.5.

**Theorem 5.7.** *Under Assumption 5.6, for any $\varepsilon > 0$, $\alpha \in [0, 1/2]$, $\delta \in (0,1)$, and $n \geq \frac{8\ln(d/\delta)}{\xi}$, with probability at least $1-\delta$, Algorithm 1 under CTL and LTC achieves*

$$\left\|\widehat{\theta}_{\text{CTL}} - \theta_{\text{true}}\right\|_2 \leq C\left(\frac{\alpha}{\gamma\xi} + \frac{c(\varepsilon)}{\gamma\xi}\sqrt{\frac{\ln\frac{1}{\delta}}{n}}\right),$$

$$\left\|\widehat{\theta}_{\text{LTC}} - \theta_{\text{true}}\right\|_2 \leq C\left(\frac{c(\varepsilon)\alpha}{\gamma\xi} + \frac{c(\varepsilon)}{\gamma\xi}\sqrt{\frac{\ln\frac{1}{\delta}}{n}}\right).$$

Here, we see that the separation between CTL and LTC still

exists, with an additional factor of $c(\varepsilon)$ in LTC, illustrating a negative impact of LDP on robustness.

# 6. Putting It All Together: Suboptimality under RLHF and DPO

In this section, we are ready to present our main results on the suboptimality gap under RLHF and DPO by combining our reduction results with estimation error bounds.

## 6.1. Private and Robust RLHF

**Theorem 6.1.** *Under the conditions of Corollary 4.4 and Theorem 5.4, RLHF (Algorithm 2) achieves the following suboptimality with probability at least $1 - \delta$*

$$\text{SubOpt}_{\text{CTL}}(\widehat{\pi}, \pi^\dagger) \leq C\sqrt{d \cdot \kappa(\pi^\dagger, \pi_{\text{ref}})}$$
$$\times \left( \frac{\sqrt{\alpha}}{\gamma} + \frac{c(\varepsilon)}{\gamma}\sqrt{\frac{d + \ln\frac{1}{\delta}}{n}} + B\sqrt{\lambda} \right),$$

$$\text{SubOpt}_{\text{LTC}}(\widehat{\pi}, \pi^\dagger) \leq C\sqrt{d \cdot \kappa(\pi^\dagger, \pi_{\text{ref}})}$$
$$\times \left( \frac{c(\varepsilon)\sqrt{\alpha}}{\gamma} + \frac{c(\varepsilon)}{\gamma}\sqrt{\frac{d + \ln\frac{1}{\delta}}{n}} + B\sqrt{\lambda} \right),$$

*for any comparator policy $\pi^\dagger$ and $\lambda \geq \Omega\left(\frac{d}{n} \cdot \ln(n/\delta)\right)$.*

The proof follows directly from the reduction result in Corollary 4.4 and estimation error bound in Theorem 5.4. To the best of our knowledge, this is the first result on the suboptimality performance of RLHF under both privacy and corruption. In particular, let $\lambda = \widetilde{\Theta}(d/(B^2\gamma^2 n)) \geq \widetilde{\Omega}(d/n)$, the sample complexity part in the bounds (i.e., the last two terms) approaches zero with a rate of $\widetilde{\mathcal{O}}(\sqrt{d/n})$, but with a multiplicative factor of $c(\varepsilon)$ that captures the cost of privacy. Meanwhile, due to strong corruption, a non-vanishing bias term exists in all three cases in terms of corruption parameters, which illustrates an interesting interplay between privacy and robustness, discussed below.

**Separation between CTL and LTC.** One key observation is that LDP before corruption leads to an additional $c(\varepsilon)$ factor in the bias term, which mimics the same phenomena in private and robust mean estimation problems (Zhou & Zhang, 2024; Cheu et al., 2021).

**Comparisons with Prior Work.** We now highlight our contributions even in robust-only or private-only RLHF, by comparing our result above with existing ones where privacy and robustness are separately considered.

**1. Robust RLHF:** To our best knowledge, only recent work (Mandal et al., 2024) establishes theoretical suboptimality bounds for RLHF under adversarial corruption. In particular, it takes a linear MDP view (rather than our linear bandit view) of RLHF under strong corruption of both features and labels. Under the same *relative condition number* assumption, their dependence on $\alpha$ is $\mathcal{O}(\alpha^{1/4})$ when reduced from MDP to bandit. In contrast, our result gives a better dependence $\mathcal{O}(\sqrt{\alpha})$, although only with label corruption. It is worth noting that this $\mathcal{O}(\sqrt{\alpha})$ dependence is state-of-the-art even in the easier setting of standard offline reinforcement learning (Zhang et al., 2022). Moreover, our Algorithm 1 is much simpler than the one in (Mandal et al., 2024). Thus, a fair conclusion here could be that our result offers a better algorithm and theoretical result in the easier label-only corruption setting.

**2. Private RLHF:** To our best knowledge, we are unaware of prior work that explicitly states the private suboptimality of RLHF in terms of relative condition number, often used in the standard offline RL. The most related one is Chowdhury et al. (2023), which generalizes the non-private RLHF in Zhu et al. (2023) to the same locally private one as ours. However, both Chowdhury et al. (2023) and Zhu et al. (2023) state their suboptimality as

$$\text{SubOpt}(\widehat{\pi}, \pi^\star) \leq \|\mathbb{E}_{s\sim\rho}[\phi(s, \pi^\star(s)) - v]\|_{(\widehat{\Sigma}+\lambda I)^{-1}}$$
$$\times 2F(n, d, \delta, \lambda), \quad (15)$$

for any chosen reference vector $v \in \mathbb{R}^d$ and some function $F$. This is similar to our intermediate result in (6) but has some key differences. One potential issue in (15) is that it does not offer clear guidance on choosing the important vector $v$. In particular, if $v = 0$, then the suboptimality may not converge to zero as $n \to \infty$. This is because in both papers, $\lambda$ has to be on the order of $1/n$ so as to ensure that $F(n, d, \delta, \lambda) \leq \mathcal{O}(1/\sqrt{n})$. However, in this case, if the minimum eigenvalue of the empirical matrix $\widehat{\Sigma}$ is small, the norm term $\|\mathbb{E}_{s\sim\rho}[\phi(s, \pi^\star(s)) - v]\|_{(\widehat{\Sigma}+\lambda I)^{-1}}$ can be on the order of $\sqrt{n}$, given the choice of $\lambda$. To partially address this, Zhu et al. (2023) suggest a heuristic way of selecting $v$ as the most common feature vector that appears in the data set. In contrast, we consider a reference policy $\pi_{\text{ref}}$ and offer a theory-grounded rule for selecting it via relative condition number along with Corollary 4.4.

Our next result is the suboptimality in RLHF under the assumption of uniform coverage (cf. Assumption 5.6).

**Theorem 6.2.** *Under the conditions of Proposition 4.2 and for $n \geq \frac{8\ln(d/\delta)}{\xi}$, RLHF (Algorithm 2) achieves the following suboptimality with probability at least $1 - \delta$*

$$\text{SubOpt}_{\text{CTL}}(\widehat{\pi}, \pi^\star) \leq C\left( \frac{\alpha}{\gamma\xi} + \frac{c(\varepsilon)}{\gamma\xi}\sqrt{\frac{\ln\frac{1}{\delta}}{n}} \right),$$

$$\text{SubOpt}_{\text{LTC}}(\widehat{\pi}, \pi^\star) \leq C\left( \frac{c(\varepsilon)\alpha}{\gamma\xi} + \frac{c(\varepsilon)}{\gamma\xi}\sqrt{\frac{\ln\frac{1}{\delta}}{n}} \right).$$

The proof follows directly from Proposition 4.2 and Theorem 5.7. Compared with Theorem 6.1, the corruption term becomes $\alpha$ (with a factor of $1/\xi$) rather than $\sqrt{\alpha}$ while the concentration part has no explicit dependence on $d$ but with $1/\xi$ factor, which however implicitly depends on $d$. As before, a separation exists between CTL and LTC, due to the additional $c(\varepsilon)$ factor in LTC. It is worth noting that the $\mathcal{O}(\alpha/\xi)$ dependence matches the best existing result in standard offline RL under corruption in Zhang et al. (2022).

**Comparisons with Prior Work.** Mandal et al. (2024) also consider the uniform coverage case and establish a bias corruption term on the order of $\frac{\sqrt{d}\alpha^{1-o(1)}}{\xi}$ when reduced from their MDP to bandit setting. In contrast, in our label-corruption setting, we have no *explicit* dependence on $d$ and a better dependence on $\alpha$. Moreover, we highlight that the missing dependence of $1/\gamma$ in Mandal et al. (2024) is actually due to an error in their proof (see Appendix G for a detailed discussion). That is, the correct bound of their algorithm also has a $1/\gamma$ factor. In the context of private RLHF under uniform coverage, our bound matches the state-of-the-art in Chowdhury et al. (2023) when the corruption parameter is zero.

### 6.2. Private and Robust DPO

Thanks to our reduction result, we can also leverage the estimation error bound to give the *first* result on suboptimality in DPO-style algorithms under privacy and corruption.

**Theorem 6.3.** *Under the conditions of Proposition 4.6, the policy corresponding to the output of Algorithm 1 achieves the following suboptimality with probability at least $1 - \delta$*

$$\text{SubOpt}_{\text{CTL}}(\widehat{\pi}, \pi^\star) \leq C \cdot B\sqrt{\kappa_\Pi}$$
$$\times \left( \frac{\sqrt{\alpha}}{\gamma} + \frac{c(\varepsilon)}{\gamma}\sqrt{\frac{d + \ln\frac{1}{\delta}}{n}} + \beta B\sqrt{\lambda} \right),$$

$$\text{SubOpt}_{\text{LTC}}(\widehat{\pi}, \pi^\star) \leq C \cdot B\sqrt{\kappa_\Pi}$$
$$\times \left( \frac{c(\varepsilon)\sqrt{\alpha}}{\gamma} + \frac{c(\varepsilon)}{\gamma}\sqrt{\frac{d + \ln\frac{1}{\delta}}{n}} + \beta B\sqrt{\lambda} \right),$$

*for $\beta > 0$, $\lambda \geq \Omega\left(\frac{d}{n} \cdot \ln(n/\delta)\right)$, $\gamma = 1/(2 + \exp(-\beta B) + \exp(\beta B))$, and some universal constant $C > 0$.*

*Remark* 6.4. The policy in the above theorem in fact corresponds to the output of the algorithm rDPO proposed in Chowdhury et al. (2024) with a log-linear policy class, see Appendix C for more details. That is, while Chowdhury et al. (2024) only shows a suboptimal rate for rDPO, we are the first to attain $O(1/\sqrt{n})$ rate, see more discussion below.

The proof follows from Proposition 4.6 and Theorem 5.4 with $B' = \mathcal{O}(\beta B)$. To our knowledge, this is the first theoretical result on DPO-style algorithms under privacy and

corruption. As before, we can see that the interplay of local privacy and adversarial corruption introduces a separation between CTL and LTC by a factor of $c(\varepsilon)$. Moreover, our result also significantly advances the state-of-the-art for DPO-style algorithms under privacy or corruption separately, as discussed in detail below.

**Private DPO.** Consider $\alpha = 0$, $\lambda = \widetilde{\Theta}(d/(\beta^2 B^2 \gamma^2 n)) \geq \widetilde{\Omega}(d/n)$, we obtain the first suboptimality for private DPO with rate $\widetilde{O}(1/\gamma \cdot c(\varepsilon)\sqrt{d/n} \cdot \sqrt{\kappa_\Pi})$, where $c(\varepsilon)$ is the additional cost due to local privacy. The rate matches the best possible non-private one as $\varepsilon \to \infty$ (Song et al., 2024).

**Robust DPO.** To the best of our knowledge, only the recent work by Chowdhury et al. (2024) provides a formal theoretical bound on the suboptimality of rDPO under label corruption. In particular, it considers the random-flipping corruption model (i.e., with some *known* probability, the true label is flipped). This is a much weaker model than ours and, in fact, is equivalent to local privacy after re-parameterization. Under this weaker model, Chowdhury et al. (2024) only established a suboptimal rate of $\widetilde{\mathcal{O}}(1/n^{1/4})$ in the general case, while our result implies a rate of $\widetilde{\mathcal{O}}(1/n^{1/2})$ (by using our private DPO result above) under the same corruption model. Moreover, moving from this weaker corruption model to a corruption model in the robust statistics literature (i.e., strong corruption model), our result above shows that rDPO now suffers a non-vanishing bias term.

**Practical Implementation and Experiments.** Given that Theorem 6.3 establishes the SOTA theoretical results of rDPO in both private and corruption cases, under the log-linear policy. One may also interested in its empirical performance in general with neural nets as the policy class. We have a series of experiments (see Appendix D for details), which demonstrate some interesting results.

## 7. Discussion and Conclusion

While we present only upper bound results in the main body, we briefly discuss their tightness here; for further details, please refer to Appendix C. First, when $\alpha = 0$, the additional factor $c(\varepsilon)$ due to privacy matches the minimax lower bound established in Chowdhury et al. (2023). Furthermore, the dependence on $1/\gamma = \Theta(e^B) = \Theta(e^{r_{\max}})$ appears in nearly all existing results on both offline and online RLHF (Zhu et al., 2023; Zhan et al., 2023; Xie et al., 2024; Pacchiano et al., 2021; Chen et al., 2022), stemming from the non-linearity of the Bradley-Terry model. Second, in the limit $\varepsilon \to \infty$ (non-private case), our dependence on $\alpha$ is $\mathcal{O}(\sqrt{\alpha})$ and $\mathcal{O}(\alpha/\zeta)$ (under uniform coverage), both of which align with state-of-the-art results in standard offline RL settings, where rewards rather than preferences are observed. In fact, we conjecture that the $\mathcal{O}(\alpha/\zeta)$ dependence is optimal. Third, regarding the separation between CTL

and LTC, the conclusion is nuanced. We tend to believe that the additional factor $c(\varepsilon)$ in the uniform coverage case is tight, as it matches the known result in mean estimation and offline bandits (Zhou & Zhang, 2024). However, under the $\mathcal{O}(\sqrt{\alpha})$ dependence without coverage, we hypothesize that achieving an $\mathcal{O}(\sqrt{c(\varepsilon)})$ separation—rather than $\mathcal{O}(c(\varepsilon))$— is possible, presenting an exciting direction for future work. Looking ahead, our reduction analysis and new results on private and robust alignment may serve as key benchmarks and inspire further research in this domain.

## Acknowledgements

XZ is supported in part by NSF CNS-2153220 and CNS-2312835. XZ would like to thank Weihao Kong for insightful discussions.

## Impact Statement

This paper presents work whose goal is to advance the field of Machine Learning. There are many potential societal consequences of our work, none which we feel must be specifically highlighted here.

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

# A. Additional Related Work

We discuss here more relevant work that do not fit in the main text. In addition to the work discussed below, we refer readers to Huang et al. (2024) for theoretical results on standard offline alignment, to the survey Casper et al. (2023) for a more comprehensive overview of RLHF, and to Wang et al. (2024) for the overview of LLM alignment in general.

**Provably robust alignment under corruption.** We would like to remark that our use of the strong corruption model from robust statistics literature is motivated by its popularity in robust offline and online reinforcement learning (i.e., when the actual rewards are observed) (Zhang et al., 2022; 2021), as well as the recent interest in examining its interplay with local differential privacy across various statistical tasks (Li et al., 2023; Cheu et al., 2021; Chhor & Sentenac, 2023). Moreover, this corruption model allows us to consider corruption occurring in both data generation and collection.

**Provably robust offline RL.** Without privacy constraints, our work can be seen as a non-trivial extension of the results in corruption-robust offline RL (Zhang et al., 2022) to the setting of offline RLHF, where only relative rankings, rather than true rewards, are observed. As will be discussed in Appendix C, the lower bounds established for robust offline RL, along with their proof techniques, can be applied or adapted to derive lower bounds for offline RLHF.

**Robust logistic regression under corruption.** Among those works on logistic regression under adversary corruption (Feng et al., 2014; Prasad et al., 2020; Chen et al., 2020; Awasthi et al., 2022), the most relevant one is Awasthi et al. (2022) that considers Binomial regression under label corruption, which includes logistic regression as a special case. Awasthi et al. (2022) propose an alternating minimization method that achieves a recover rate of $\mathcal{O}(\alpha \ln(1/\alpha))$ in $L_2$ norm, where $\alpha \in [0, 1/2)$ is the corruption parameter. In contrast, our intermediate result in Section 5 implies a rate of $\mathcal{O}(\alpha)$. Moreover, our rate is achieved by the simple maximum likelihood estimator rather than the inefficient trimmed maximum likelihood estimator in Awasthi et al. (2022).

# B. Algorithm

---

**Algorithm 2** Offline RLHF

---

1: **Input:** The current parameter estimate $\widehat{\theta}$, the empirical covariance matrix $\widehat{\Sigma}$, the regularizer $\lambda$, the concentration bound $\Gamma(n, d, \delta, \lambda)$, a reference policy $\pi_{\mathrm{ref}}$ and a tuning parameter $\eta \in \{0, 1\}$.
2: **if** $\eta = 0$ **then**
3:     $\widehat{J}(\pi) = \mathbb{E}_{s \sim \rho, a \sim \pi(\cdot|s)}[\langle \widehat{\theta}, \phi(s, a) \rangle]$
4:     **return** $\widehat{\pi} = \mathrm{argmax}_{\pi} \widehat{J}(\pi)$
5: **else**
6:     Construct confidence set

$$\Theta(\widehat{\theta}, \lambda) = \left\{ \theta \in \Theta_B \mid \|\widehat{\theta} - \theta\|_{\widehat{\Sigma} + \lambda I} \leq \Gamma(n, d, \delta, \lambda) \right\}$$

    Compute pessimistic expected value

$$\widehat{J}(\pi) = \min_{\theta \in \Theta(\widehat{\theta}, \lambda)} \mathbb{E}_{s \sim \rho, a \sim \pi(\cdot|s)}[\langle \theta, \phi(s, a) \rangle] - \mathbb{E}_{s \sim \rho, a \sim \pi_{\mathrm{ref}}(\cdot|s)}[\langle \theta, \phi(s, a) \rangle]$$

7:     **return** $\widehat{\pi} = \mathrm{argmax}_{\pi} \widehat{J}(\pi)$
8: **end if**

---

# C. Discussions

In this section, we discuss the tightness of our suboptimality bounds. In particular, we primarily focus on the result in Theorem 6.1, as it offers stronger guarantees compared to Theorem 6.3.

**Dependence on $1/\gamma$.** The dependence on $1/\gamma = \Theta(e^B) = \Theta(e^{r_{\max}})$ is present in nearly all existing results on both offline and online RLHF (Zhu et al., 2023; Zhan et al., 2023; Xie et al., 2024; Pacchiano et al., 2021; Chen et al., 2022). This stems from an intrinsic feature of the Bradley-Terry model, namely, the non-linearity of the sigmoid function.

**The privacy cost of $c(\varepsilon)$.** Compared to the non-private (non-corrupted) case, our bound includes an additional multiplicative

factor of $c(\varepsilon)$, which we believe to be tight when $\varepsilon \in [0, 1]$, i.e., $c(\varepsilon) = \Theta(1/\varepsilon)$. First, this factor appears even in simple mean estimation, where a matching lower bound is provided in Duchi et al. (2018). Second, a more concrete argument can be made by modifying the existing lower bound proof for the non-private case to show that $c(\varepsilon)$ is necessary. Specifically, the key insight is that any LDP mechanism is a contraction of the KL divergence, as stated in Duchi et al. (2018, Theorem 1). Thus, in the lower bound proof for the private case, the non-private KL divergence is replaced with the private one, which is smaller by a factor of $(e^{\varepsilon} - 1)^2$, eventually leading to a factor of $1/\varepsilon$.

**The separation between CTL and LTC.** We observe an additional factor of $c(\varepsilon)$ in the corruption under LTC compared to CTL. We conjecture that this is tight for all $\varepsilon > 0$, especially for the one in Theorem 6.2. First, the separation result is also seen in the mean estimation problem and is shown to be tight (Zhou & Zhang, 2024). Second, a more concrete argument can be made by modifying the lower bound for standard offline linear bandits under corruption (Zhang et al., 2022).[1] This lower bound is valid for offline RLHF under CTL,[2] as offline RLHF is at least as hard as offline linear bandits, and CTL is harder than corruption-only settings. To demonstrate the additional $c(\varepsilon)$ factor under LTC, a key fact is that any LDP mechanism contracts the total variation distance by a factor of $c(\varepsilon)$ (cf. Lemma H.4). Using a standard coupling argument, one can then derive a lower bound with the additional factor of $c(\varepsilon)$ for the LTC setting.

**Dependence on $\alpha$.** Our current $\sqrt{\alpha}$ dependence matches the best existing result, even in standard offline RL (Zhang et al., 2022). However, this $\sqrt{\alpha}$ dependence does not align with the existing $\Omega(\alpha)$ lower bound (Zhang et al., 2022). On the other hand, under the uniform coverage assumption, our result in Theorem 6.2 achieves the optimal dependence on $\alpha$. Furthermore, we conjecture that the $1/\xi$ factor preceding $\alpha$ is optimal. Our reasoning is as follows: due to boundedness, we have $\xi \le 1/d$. In the best case, when $\xi = 1/d$, our upper bound matches the lower bound of $d\alpha$ in Zhang et al. (2022), which was established for standard offline linear RL, except for the difference of $1/\gamma$ due to the non-linearity.

**Practical implementations.** For the sake of theoretical analysis, we adopt linear modeling in the main paper. Nevertheless, we mention that our proposed method can be readily extended to the case with general function classes (albeit losing the current formal theoretical guarantees). Take DPO for an example, we can solve the following optimization problem:

$$\widehat{\pi} = \underset{\pi \in \Pi}{\arg\min} \ -\left( \sum_{i=1}^{n} \ln(1 - \sigma(r_{\beta,i}^{\pi,\pi_{\text{sft}}})) + (z_i + \sigma(\varepsilon) - 1)c(\varepsilon) \ln\left( \frac{\sigma(r_{\beta,i}^{\pi,\pi_{\text{sft}}})}{1 - \sigma(r_{\beta,i}^{\pi,\pi_{\text{sft}}})} \right) \right), \tag{16}$$

where

$$r_{\beta,i}^{\pi,\pi_{\text{sft}}} := \beta \ln \frac{\pi(a_i^1|s_i)}{\pi_{\text{sft}}(a_i^1|s_i)} - \beta \ln \frac{\pi(a_i^0|s_i)}{\pi_{\text{sft}}(a_i^0|s_i)} \ .$$

Some sanity checks are in order. First, for the standard case (i.e., $\varepsilon \to \infty$ and $\alpha = 0$), we have $\sigma(\varepsilon) = c(\varepsilon) = 1$ and $z_i = y_i$, which leads us back to the standard DPO loss, see (11). Second, if we consider log-linear policy, (16) reduces to (14) (up to some scaling of $\beta$). Third, if there is only privacy (or similar random flipping noise with a known flipping rate as in Chowdhury et al. (2024)), one can verify that the above loss is equivalent to the one in Chowdhury et al. (2024) (see their Eq. 12, which is called rDPO), up to some simple rescaling. Thus, in this sense, compared to the sub-optimal rate of $\mathcal{O}(1/n^{1/4})$ for the log-linear policy class established in Chowdhury et al. (2024), we give the first $\mathcal{O}(1/\sqrt{n})$ rate for private or "robust" DPO. One can also follow a similar approach as above by simply replacing the policy-parameterized reward $r_{\beta,i}^{\pi,\pi_{\text{sft}}}$ by a reward function in a reward function class for RLHF. Then, a similar method as in Algorithm 1 of Zhan et al. (2023) can be adopted for introducing pessimism.

## D. Experiments on DPO and rDPO under Privacy and Corruption

As mentioned in the last section, we provide the first results for rDPO (Chowdhury et al., 2024) under both privacy and corruption with a log-linear policy class (cf. Theorem 6.3). In this section, we would like to empirically demonstrate its performance with a general function class, i.e., neural nets.

---

[1]Note that the hard instance in Zhang et al. (2022) only requires corruption in rewards, not both features and rewards. Hence, it can be used for our setting.

[2]With a factor of two in the sample complexity.

### D.1. Experiment Setup

**Dataset.** We utilize GPT-4o to generate a synthetic dataset, referred to as `finance_preference`, which comprises 1697 preference samples. Each sample includes a prompt related to a financial scenario and two possible responses, where "rejected" represents the high-risk option and "chosen" represents the low-risk option. This labeling can be viewed as private or sensitive information. For illustrative examples from our dataset, please refer to Appendix I. For SFT training, we construct the `finance_sft` dataset by simply concatenating the prompt with the corresponding "chosen" response.

**SFT Training.** We begin by fine-tuning GPT2-large using the `finance_sft` dataset to obtain the SFT policy, $\pi_{\text{sft}}$. For this, we directly utilize the SFT trainer from the Transformer Reinforcement Learning (TRL) library (von Werra et al., 2020), with the hyperparameters listed in Table 3.

**DPO and rDPO Training.** For alignment training, we split the dataset into $85\%$ for training, $5\%$ for validation, and $10\%$ for testing. For DPO, we utilize the implementation provided in the TRL library, using the hyperparameters listed in Table 4. Similarly, for rDPO, we leverage the TRL implementation, which corresponds to DPO with `lose_type` set to "robust." In the private setting with a privacy budget of $\varepsilon$, one can simply set `label_smoothing` to the flip rate, given by $\frac{1}{e^{\varepsilon}+1}$. This setting recovers the same algorithm presented in our main paper when the policy class is log-linear. Finally, we use the same set of hyperparameters for rDPO as in DPO training.

**CTL and LTC Settings.** The LDP mechanism follows the randomized response model, where the flip rate is given by $\frac{1}{e^{\varepsilon}+1}$. For corruption, we assume that a randomly sampled subset of $O(\alpha n)$ labels are always flipped compared to the true label. To implement both privacy and corruption, we introduce a mask variable initialized to 0 for each sample. The LDP mechanism flips the mask variable with probability $\frac{1}{e^{\varepsilon}+1}$, while the corruption mechanism sets the mask to 1 with probability $\alpha$. Finally, after CTL or LTC processing, labels ("chosen" and "rejected") are flipped if the corresponding mask value is 1. At this point, an astute reader may notice that LTC results in a higher number of 1s in the final mask variables compared to CTL

**Evaluation.** We evaluate our trained models $\pi_{\text{DPO}}$, $\pi_{\text{rDPO}}$, and $\pi_{\text{SFT}}$ by generating responses for the test dataset using the hyperparameters listed in Table 5. To assess performance, we employ the `llama3:70b` model as a judge, comparing responses from $\pi_{\text{DPO}}$ and $\pi_{\text{rDPO}}$ against those from $\pi_{\text{SFT}}$. Finally, we use the win rate from these comparisons as our primary performance metric, following the methodology outlined in the DPO paper (Rafailov et al., 2023). We compute the average and standard deviation across five seeds.

### D.2. Results

**Private Case.** We first compare the performance of DPO and rDPO in the private setting, as shown in Table 1. Due to the "shifting and scaling" loss used in rDPO, we observe that rDPO outperforms standard DPO in the private case. Interestingly, we make an additional observation: in the non-private setting, if we still introduce random label flips at a rate of approximately $1/(e^1+1)$, rDPO achieves even better performance than DPO. This suggests that deliberately adding noise to labels can enhance performance, resembling the well-known effect of label smoothing in classification tasks. We also tend to believe that this injected noise also somewhat help to address the overoptimization issues in DPO-style algorithms. We plan to further explore this phenomenon on a larger dataset. Finally, we note that this observation does not contradict our main theoretical result, which provides an upper bound in the worst case.

**Private and Corruption Cases.** We now examine whether the separation between CTL and LTC persists beyond the linear setting. As shown in Table 2, rDPO demonstrates better performance under CTL compared to LTC. Furthermore, the performance gap widens as $\varepsilon$ decreases. These observations are consistent with the theoretical insights derived from the linear setting.

*Table 1.* Comparison of win rates (%) for DPO and rDPO across different values of privacy budget $\varepsilon$.

| $\varepsilon$ | rDPO winrate (%) | DPO winrate (%) |
|---|---|---|
| 0.1 | **59.0 $\pm$ 4.7** | 55.4 $\pm$ 1.1 |
| 0.5 | **65.8 $\pm$ 5.6** | 60.4 $\pm$ 3.0 |

*Table 2.* Comparison of win rates (%) for rDPO under CTL and LTC.

| $(\varepsilon, \alpha)$ | win rates (%) under CTL | win rates (%) under LTC |
|---|---|---|
| $(1, 0.1)$ | **69.6 ± 5.1** | 65.4 ± 5.0 |
| $(0.5, 0.1)$ | **64.4 ± 2.8** | 58.6 ± 2.6 |

# E. Proofs

This section presents the proofs for our main results in previous sections.

## E.1. Proof of Proposition 4.2

*Proof.* By definition, for any $\pi^\dagger$, we have

$$\mathrm{SubOpt}(\widehat{\pi}, \pi^\dagger) = J(\pi^\dagger) - J(\widehat{\pi})$$
$$= \underbrace{J(\pi^\dagger) - \widehat{J}(\pi^\dagger)}_{\mathcal{T}_1} + \underbrace{\widehat{J}(\pi^\dagger) - \widehat{J}(\widehat{\pi})}_{\mathcal{T}_2} + \underbrace{\widehat{J}(\widehat{\pi}) - J(\widehat{\pi})}_{\mathcal{T}_3},$$

holds for any function $\widehat{J}(\cdot)$. For the first case when $\eta = 0$, we have $\widehat{J}(\pi) = \mathbb{E}_{s \sim \rho, a \sim \pi(\cdot|s)}[\phi(s,a)^\top \widehat{\theta}]$. By the greedy algorithm in Algorithm 2, we have $\mathcal{T}_2 \leq 0$. Further, under Assumption 4.1, we can rewrite $\mathcal{T}_1$ and $\mathcal{T}_3$ as

$$\mathcal{T}_1 = \mathbb{E}_{s \sim \rho, a \sim \pi^\dagger(\cdot|s)}[\phi(s,a)^\top (\theta^\star - \widehat{\theta})], \quad \mathcal{T}_3 = \mathbb{E}_{s \sim \rho, a \sim \widehat{\pi}(\cdot|s)}[\phi(s,a)^\top (\widehat{\theta} - \theta^\star)] .$$

By the boundedness assumption, both terms can be upper bounded by $\left\|\widehat{\theta} - \theta^\star\right\|_2$, which implies the first result by the fact that $\theta_{\mathrm{true}} = \theta^\star$.

For the second case when $\eta = 1$, we introduce the following notation

$$J(\pi; \theta^\star) := \mathbb{E}_{s \sim \rho, a \sim \pi(\cdot|s)}[\phi(s,a)^\top \theta^\star] = J(\pi) . \tag{17}$$

Thus, we have $\mathbb{E}_{s \sim \rho, a \sim \pi(\cdot|s)}[\langle \theta, \phi(s,a) \rangle] - \mathbb{E}_{s \sim \rho, a \sim \pi_{\mathrm{ref}}(\cdot|s)}[\langle \theta, \phi(s,a) \rangle] = J(\pi; \theta) - J(\pi_{\mathrm{ref}}; \theta)$. Let $\theta_\pi^{\mathrm{inf}} = \mathrm{argmin}_{\theta \in \Theta(\widehat{\theta}, \lambda)} J(\pi; \theta) - J(\pi_{\mathrm{ref}}; \theta)$. Hence $\widehat{J}(\pi) = J(\pi; \theta_\pi^{\mathrm{inf}}) - J(\pi_{\mathrm{ref}}; \theta_\pi^{\mathrm{inf}})$. Then, we have

$$\mathrm{SubOpt}(\widehat{\pi}, \pi^\dagger) = J(\pi^\dagger) - J(\widehat{\pi})$$
$$= J(\pi^\dagger; \theta^\star) - J(\pi_{\mathrm{ref}}; \theta^\star) - (J(\widehat{\pi}; \theta^\star) - J(\pi_{\mathrm{ref}}; \theta^\star))$$
$$\overset{(a)}{\leq} \left(J(\pi^\dagger; \theta^\star) - J(\pi_{\mathrm{ref}}; \theta^\star)\right) - \left(J(\pi^\dagger; \theta_{\pi^\dagger}^{\mathrm{inf}}) - J(\pi_{\mathrm{ref}}; \theta_{\pi^\dagger}^{\mathrm{inf}})\right)$$
$$+ \left(J(\widehat{\pi}; \theta_{\widehat{\pi}}^{\mathrm{inf}}) - J(\pi_{\mathrm{ref}}; \theta_{\widehat{\pi}}^{\mathrm{inf}})\right) - (J(\widehat{\pi}; \theta^\star) - J(\pi_{\mathrm{ref}}; \theta^\star))$$
$$\overset{(b)}{\leq} \left(J(\pi^\dagger; \theta^\star) - J(\pi_{\mathrm{ref}}; \theta^\star)\right) - \left(J(\pi^\dagger; \theta_{\pi^\dagger}^{\mathrm{inf}}) - J(\pi_{\mathrm{ref}}; \theta_{\pi^\dagger}^{\mathrm{inf}})\right)$$
$$= \underbrace{\left(J(\pi^\dagger; \theta^\star) - J(\pi_{\mathrm{ref}}; \theta^\star)\right) - \left(J(\pi^\dagger; \widehat{\theta}) - J(\pi_{\mathrm{ref}}; \widehat{\theta})\right)}_{\mathcal{T}_4}$$
$$+ \underbrace{\left(J(\pi^\dagger; \widehat{\theta}) - J(\pi_{\mathrm{ref}}; \widehat{\theta})\right) - \left(J(\pi^\dagger; \theta_{\pi^\dagger}^{\mathrm{inf}}) - J(\pi_{\mathrm{ref}}; \theta_{\pi^\dagger}^{\mathrm{inf}})\right)}_{\mathcal{T}_5},$$

where $(a)$ holds by the greedy algorithm; $(b)$ holds by the definition of $\theta_{\widehat{\pi}}^{\mathrm{inf}}$ and the fact that $\theta^\star \in \Theta(\widehat{\theta}, \lambda)$ by (5). To bound $\mathcal{T}_4$ and $\mathcal{T}_5$, we use the definition in (17), the concentration in (5) and the definition of $\Theta(\widehat{\theta}, \lambda)$ with $\theta_{\pi^\dagger}^{\mathrm{inf}} \in \Theta(\widehat{\theta}, \lambda)$, and obtain that

$$\mathcal{T}_4 + \mathcal{T}_5 \leq 2\Gamma(n, d, \delta, \lambda) \left\|\mathbb{E}_{s \sim \rho}[\phi(s, \pi^\dagger(s)) - \phi(s, \pi_{\mathrm{ref}}(s))]\right\|_{(\widehat{\Sigma} + \lambda I)^{-1}},$$

where we let $\phi(s, \pi(s)) := \mathbb{E}_{a \sim \pi(\cdot|s)}[\phi(s,a)]$. This finishes the proof. $\square$

## E.2. Proof of Corollary 4.4

*Proof.* We need only focus on the last term in (6). Note that

$$\mathbb{E}_{s\sim\rho}[\phi(s,\pi^{\dagger}(s)) - \phi(s,\pi_{\mathrm{ref}}(s))] = \mathbb{E}_{s\sim\rho,a\sim\pi^{\dagger}(\cdot|s),a'\sim\pi_{\mathrm{ref}}(\cdot|s)}[\phi(s,a) - \phi(s,a')].$$

Thus, we have

$$\left\|\mathbb{E}_{s\sim\rho}[\phi(s,\pi^{\dagger}(s)) - \phi(s,\pi_{\mathrm{ref}}(s))]\right\|^{2}_{(\widehat{\Sigma}+\lambda I)^{-1}}$$

$$= \left\|\mathbb{E}_{s\sim\rho,a\sim\pi^{\dagger}(\cdot|s),a'\sim\pi_{\mathrm{ref}}(\cdot|s)}[\phi(s,a) - \phi(s,a')]\right\|^{2}_{(\widehat{\Sigma}+\lambda I)^{-1}}$$

$$\overset{(a)}{\leq} 3\left\|\mathbb{E}_{s\sim\rho,a\sim\pi^{\dagger}(\cdot|s),a'\sim\pi_{\mathrm{ref}}(\cdot|s)}[\phi(s,a) - \phi(s,a')]\right\|^{2}_{(\Sigma^{\mathrm{diff}}_{\pi_{\mathrm{sft}},\pi_{\mathrm{sft}}}+\lambda I)^{-1}}$$

$$\overset{(b)}{\leq} 3\cdot\kappa(\pi^{\dagger},\pi_{\mathrm{ref}})\cdot\left\|\mathbb{E}_{s\sim\rho,a\sim\pi^{\dagger}(\cdot|s),a'\sim\pi_{\mathrm{ref}}(\cdot|s)}[\phi(s,a) - \phi(s,a')]\right\|^{2}_{\left(\Sigma^{\mathrm{diff}}_{\pi^{\dagger},\pi_{\mathrm{ref}}}\right)^{-1}}$$

$$\overset{(c)}{\leq} 3\cdot\kappa(\pi^{\dagger},\pi_{\mathrm{ref}})\cdot\mathbb{E}_{s\sim\rho,a\sim\pi^{\dagger}(\cdot|s),a'\sim\pi_{\mathrm{ref}}(\cdot|s)}\left[(\phi(s,a) - \phi(s,a'))^{\top}\left(\Sigma^{\mathrm{diff}}_{\pi^{\dagger},\pi_{\mathrm{ref}}}\right)^{-1}(\phi(s,a) - \phi(s,a'))\right]$$

$$\overset{(d)}{=} 3\cdot\kappa(\pi^{\dagger},\pi_{\mathrm{ref}})\cdot\mathrm{trace}(I),$$

where $(a)$ holds by Lemma H.1 for $\lambda \geq \Omega\left(\frac{d}{n}\cdot\ln(n/\delta)\right)$; $(b)$ follows by the definition of $\kappa(\pi^{\dagger},\pi_{\mathrm{ref}})$ in (8); $(c)$ holds by Jensen's inequality; $(d)$ simply follows from the interchange of trace and expectation along with the cyclic property of trace. Taking the square root, yields the required result. $\qquad\square$

## E.3. Proof of Proposition 4.6

*Proof.* We first show that under Assumption 4.5, the labels are generated via a logistic regression model. This follows from a direct computation. In particular, by (1), (10), (12), we have

$$\begin{aligned}
\mathbb{P}\left(y_i = 1|s_i, a_i^0, a_i^1\right) &= \frac{1}{1 + \exp(r^{\star}(s_i, a_i^0) - r^{\star}(s_i, a_i^1))} \\
&= \sigma(r^{\star}(s_i, a_i^1) - r^{\star}(s_i, a_i^0)) \\
&= \sigma\left(\beta\ln\frac{\pi^{\star}(a_i^1|s_i)}{\pi^{\star}(a_i^0|s_i)} - \beta\ln\frac{\pi_{\mathrm{sft}}(a_i^1|s_i)}{\pi_{\mathrm{sft}}(a_i^0|s_i)}\right) \\
&= \sigma\left(\langle\beta(\theta^{\star} - \theta_{\mathrm{sft}}), \phi(s_i, a_i^1) - \phi(s_i, a_i^0)\rangle\right).
\end{aligned}$$

Thus, with $\theta_{\mathrm{true}} = \beta(\theta^{\star} - \theta_{\mathrm{sft}})$ and $x_i = \phi(s_i, a_i^1) - \phi(s_i, a_i^0)$, we have that each label $y_i$ follows from logistic regression in (3).

We now turn to the suboptimality part.

$$\begin{aligned}
\mathrm{SubOpt}(\widehat{\pi}, \pi^{\star}) &= \mathbb{E}_{s\sim\rho,a\sim\pi^{\star}}[r^{\star}(s,a)] - \mathbb{E}_{s\sim\rho,a\sim\widehat{\pi}}[r^{\star}(s,a)] \\
&\overset{(a)}{\leq} \Delta_{\max}\mathbb{E}_{s\sim\rho}\left[\mathrm{TV}(\pi^{\star}(\cdot|s),\widehat{\pi}(\cdot|s))\right] \\
&\overset{(b)}{\leq} \Delta_{\max}\mathbb{E}_{s\sim\rho}\left[\sqrt{1/2\cdot\mathrm{KL}(\pi^{\star}(\cdot|s),\widehat{\pi}(\cdot|s))}\right] \\
&\overset{(c)}{\leq} \Delta_{\max}\sqrt{1/2\cdot\mathbb{E}_{s\sim\rho}\left[\mathrm{KL}(\pi^{\star}(\cdot|s),\widehat{\pi}(\cdot|s))\right]},
\end{aligned}$$

where in $(a)$ we have $\Delta_{\max} = \max_{s,a}\left(r^{\star}(s,a) - \beta\ln Z_{\beta}(s)\right) \leq 2\beta B$, $(b)$ follows from Pinsker's inequality, and $(c)$ holds by Jensen's inequality.

Then, since both $\pi^{\star}$ and $\widehat{\pi}'$ are log-linear policies with parameters $\theta^{\star}$ and $\widehat{\theta}'$, respectively, by a direct calculation and Taylor expansion, we have

$$\mathrm{KL}(\pi^{\star}(\cdot|s),\widehat{\pi}(\cdot|s)) = \frac{1}{2}(\widehat{\theta}' - \theta^{\star})^{\top}A_s(\theta)(\widehat{\theta}' - \theta^{\star}),$$

where $A_s(\theta) := \mathbb{E}_{a \sim \pi_\theta(\cdot|s)}[\phi(s,a)\phi(s,a)^\top] - \mathbb{E}_{a \sim \pi_\theta(\cdot|s)}[\phi(s,a)]\mathbb{E}_{a \sim \pi_\theta(\cdot|s)}[\phi(s,a)]$ for some $\theta$ between $\theta^\star$ and $\widehat{\theta}'$. By independently sampling of $a, a' \sim \pi_\theta(\cdot|s)$, we have $\mathbb{E}_{s \sim \rho}[A_s(\theta)] = \frac{1}{2}\Sigma^{\text{diff}}_{\pi_\theta,\pi_\theta}$ (cf. (7)). Combing all of the above with the definition of $\kappa_\Pi$ in Definition 4.3, yields that

$$\text{SubOpt}(\widehat{\pi}, \pi^\star) \leq \sqrt{\kappa_\Pi} \cdot \frac{\Delta_{\max}}{2\sqrt{2}} \cdot \left\|\widehat{\theta}' - \theta^\star\right\|_{\Sigma^{\text{diff}}_{\pi_{\text{sft}},\pi_{\text{sft}}}}.$$

Note that $\Sigma^{\text{diff}}_{\pi_{\text{sft}},\pi_{\text{sft}}}$ is the corresponding population matrix of $\widehat{\Sigma}$. Thus, by Lemma H.1, for $\lambda \geq \Omega\left(\frac{d}{n} \cdot \ln(n/\delta)\right)$, we have

$$\text{SubOpt}(\widehat{\pi}, \pi^\star) \leq \sqrt{\kappa_\Pi} \cdot \frac{\sqrt{3}\Delta_{\max}}{2\sqrt{2}} \left\|\widehat{\theta}' - \theta^\star\right\|_{\widehat{\Sigma}+\lambda I}.$$

Finally, note that $\widehat{\theta}' - \theta^\star = (\widehat{\theta} - \theta_{\text{true}})/\beta$. Then, by (13) and $\Delta_{\max} \leq 2\beta B$, we have the final result

$$\text{SubOpt}(\widehat{\pi}, \pi^\star) \leq \frac{\sqrt{3}}{\sqrt{2}} \cdot \sqrt{\kappa_\Pi} \cdot B \cdot \Gamma(n,d,\delta,\lambda). \qquad \square$$

### E.4. Proof of Theorem 5.4

*Proof.* We divide the proof into CTL, LTC, and CLC cases. Before that, we will present some common properties of our new loss, which will be used in all three cases.

Recall that our new loss is given by

$$\widetilde{\mathcal{L}}(\theta) = -\frac{1}{n}\sum_{i=1}^{n}\widetilde{\ell}_i(\theta) \quad \text{where} \quad \widetilde{\ell}_i(\theta) = \ln(1 - \sigma(\theta^\top x_i)) + (z_i + \sigma(\varepsilon) - 1)c(\varepsilon) \cdot \theta^\top x_i,$$

where $c(\varepsilon) := \frac{1}{2\sigma(\varepsilon)-1} = \frac{e^\varepsilon+1}{e^\varepsilon-1}$. We will need its gradient and Hessian in our proof, given by

$$\nabla_\theta\widetilde{\mathcal{L}}(\theta) = -\frac{1}{n}\sum_{i=1}^{n}\left[c(\varepsilon)(z_i + \sigma(\varepsilon) - 1) - \sigma(\theta^\top x_i)\right]x_i, \tag{18}$$

$$\nabla^2_\theta\widetilde{\mathcal{L}}(\theta) = \frac{1}{n}\sum_{i=1}^{n}\left[\sigma(\theta^\top x_i)(1 - \sigma(\theta^\top x_i))\right]x_i x_i^\top, \tag{19}$$

where we use the simple fact that $\sigma'(z) = \sigma(z)(1 - \sigma(z))$.

Let $\Delta := \widehat{\theta} - \theta_{\text{true}}$, by the fact that $\widehat{\theta}$ minimizes the loss and (19), we have

$$\gamma\|\Delta\|^2_{\widehat{\Sigma}} \leq \widetilde{\mathcal{L}}(\theta_{\text{true}} + \Delta) - \widetilde{\mathcal{L}}(\theta_{\text{true}}) - \langle\nabla\widetilde{\mathcal{L}}(\theta_{\text{true}}), \Delta\rangle \leq -\langle\nabla\widetilde{\mathcal{L}}(\theta_{\text{true}}), \Delta\rangle$$
$$\leq \left\|\nabla\widetilde{\mathcal{L}}(\theta_{\text{true}})\right\|_{(\widehat{\Sigma}+\lambda I)^{-1}}\|\Delta\|_{\widehat{\Sigma}+\lambda I}, \tag{20}$$

where $\gamma = 1/(2 + \exp(-B') + \exp(B'))$ by the boundedness condition. Thus, the key is to bound the term $\left\|\nabla\widetilde{\mathcal{L}}(\theta_{\text{true}})\right\|_{(\widehat{\Sigma}+\lambda I)^{-1}}$, which will be handled separately for each case later.

For now, let us suppose we have the following high probability bound:

$$\left\|\nabla\widetilde{\mathcal{L}}(\theta_{\text{true}})\right\|_{(\widehat{\Sigma}+\lambda I)^{-1}} \leq f(n,d,\delta,\lambda), \tag{21}$$

for some function $f$, and proceed to establish the final bound. In particular, by the boundedness condition for $\Theta_{B'}$ and (20), we have

$$\gamma\|\Delta\|^2_{\widehat{\Sigma}+\lambda I} \leq \left\|\nabla\widetilde{\mathcal{L}}(\theta_{\text{true}})\right\|_{(\widehat{\Sigma}+\lambda I)^{-1}}\|\Delta\|_{\widehat{\Sigma}+\lambda I} + 4\gamma\lambda B'^2,$$

which implies that

$$\left\|\widehat{\theta} - \theta_{\text{true}}\right\|_{\widehat{\Sigma}+\lambda I} = \|\Delta\|_{\widehat{\Sigma}+\lambda I} \leq C\left(\frac{1}{\gamma} \cdot f(n,d,\delta,\lambda) + B'\sqrt{\lambda}\right), \tag{22}$$

for some universal constant $C$.

Thus, it remains to establish the high probability bound in (21) under the three settings. To this end, we will fully utilize the following claims. See Appendix F for the proofs.

**Claim E.1.** Let $\eta_i$ be zero-mean i.i.d sub-Gaussian with parameter $\sigma$, condition on $x_i$. Then, for any $\delta \in (0, 1)$ and $\lambda > 0$, with probability at least $1 - \delta$,

$$\left\| \frac{1}{n} \sum_{i=1}^{n} \eta_i x_i \right\|_{(\widehat{\Sigma} + \lambda I)^{-1}} \leq C \cdot \sigma \cdot \sqrt{\frac{d + \ln(1/\delta)}{n}},$$

for some universal constant $C$.

**Claim E.2.** Let $b = (b_1, \ldots, b_n)$ be a vector that at least $1 - \alpha n$ elements are zero, and the rest are bounded by some constant $\zeta > 0$, i.e., $|b_i| \leq \zeta$. Then, we have

$$\left\| \frac{1}{n} \sum_{i=1}^{n} b_i x_i \right\|_{(\widehat{\Sigma} + \lambda I)^{-1}} \leq \zeta \sqrt{\alpha}.$$

With the above claims in hand, we are going to establish (21) for CTL, LTC and CLC, respectively.

**CTL case.** In this case, we rewrite the gradient in (18) as follows

$$\nabla \widetilde{\mathcal{L}}(\theta_{\text{true}}) = -\frac{1}{n} \sum_{i=1}^{n} \left[ c(\varepsilon)(z_i + \sigma(\varepsilon) - 1) - \bar{y}_i + \bar{y}_i - y_i + y_i - \sigma(\theta^\top x_i) \right] x_i,$$

where recall that under CTL, the true label $y_i$ is first corrupted to $\bar{y}_i$, which will then be privatized to generate $z_i$. Thus, we have

$$\left\| \nabla \widetilde{\mathcal{L}}(\theta_{\text{true}}) \right\|_{(\widehat{\Sigma} + \lambda I)^{-1}}$$

$$\leq \underbrace{\left\| \frac{1}{n} \sum_i \left[ c(\varepsilon)(z_i + \sigma(\varepsilon) - 1) - \bar{y}_i \right] x_i \right\|_{(\widehat{\Sigma} + \lambda I)^{-1}}}_{\mathcal{T}_{\text{privacy}}} + \underbrace{\left\| \frac{1}{n} \sum_i (\bar{y}_i - y_i) x_i \right\|_{(\widehat{\Sigma} + \lambda I)^{-1}}}_{\mathcal{T}_{\text{corruption}}}$$

$$+ \underbrace{\left\| \frac{1}{n} \sum_i \left[ y_i - \sigma(\theta^\top x_i) \right] x_i \right\|_{(\widehat{\Sigma} + \lambda I)^{-1}}}_{\mathcal{T}_{\text{standard}}}. \tag{23}$$

For $\mathcal{T}_{\text{privacy}}$ and $\mathcal{T}_{\text{standard}}$, we can apply Claim E.1 due to zero-mean and sub-Gaussian with parameters $\mathcal{O}(c(\varepsilon))$ and 1, respectively. Thus, we have with probability at least $1 - \delta$,

$$\mathcal{T}_{\text{privacy}} + \mathcal{T}_{\text{standard}} \leq C_1 \cdot c(\varepsilon) \cdot \sqrt{\frac{d + \ln(1/\delta)}{n}},$$

for some universal constant $C_1 > 0$.

For $\mathcal{T}_{\text{corruption}}$, we can apply Claim E.2 with $\zeta = 1$, and obtain that

$$\mathcal{T}_{\text{corruption}} \leq \sqrt{\alpha}.$$

Thus, combining these bounds with (21) and (22), yields the bound under CTL.

**LTC case.** In this case, we rewrite the gradient in (18) as follows

$$\nabla \widetilde{\mathcal{L}}(\theta_{\text{true}}) = -\frac{1}{n} \sum_{i=1}^{n} \left[ c(\varepsilon)(z_i + \sigma(\varepsilon) - 1 + \widetilde{y}_i - \widetilde{y}_i) - \sigma(\theta^\top x_i) \right] x_i,$$

$$= -\frac{1}{n} \sum_{i=1}^{n} \left[ c(\varepsilon)(z_i - \widetilde{y}_i) + c(\varepsilon)(\widetilde{y}_i + \sigma(\varepsilon) - 1) - \sigma(\theta^\top x_i) \right] x_i,$$

where recall that under LTC, the true label $y_i$ is first privatized to be $\widetilde{y}_i$, which will then be corrupted to generate $z_i$. Thus, we have

$$
\begin{aligned}
&\left\|\nabla\widetilde{\mathcal{L}}(\theta_{\text{true}})\right\|_{(\widehat{\Sigma}+\lambda I)^{-1}} \\
&\leq \underbrace{\left\|\frac{1}{n}\sum_i\left[c(\varepsilon)(z_i-\widetilde{y}_i)\right]x_i\right\|_{(\widehat{\Sigma}+\lambda I)^{-1}}}_{\mathcal{T}_{\text{corruption}}} + \underbrace{\left\|\frac{1}{n}\sum_i\left[c(\varepsilon)(\widetilde{y}_i+\sigma(\varepsilon)-1)-\sigma(\theta^\top x_i)\right]x_i\right\|_{(\widehat{\Sigma}+\lambda I)^{-1}}}_{\mathcal{T}_{\text{privacy}}}.
\end{aligned}
\tag{24}
$$

Similarly, for $\mathcal{T}_{\text{privacy}}$, we can again apply Claim E.1 due to zero-mean and sub-Gaussian with a parameter $\mathcal{O}(c(\varepsilon))$. Thus, we have with probability at least $1-\delta$,

$$
\mathcal{T}_{\text{privacy}} \leq C_1 \cdot c(\varepsilon) \cdot \sqrt{\frac{d+\ln(1/\delta)}{n}},
$$

for some universal constant $C_1 > 0$.

For $\mathcal{T}_{\text{corruption}}$, we can apply Claim E.2 with $\zeta = c(\varepsilon)$, and obtain that

$$
\mathcal{T}_{\text{corruption}} \leq c(\varepsilon)\sqrt{\alpha}.
$$

Thus, combining these bounds with (21) and (22), yields the bound under LTC.

**CLC case.** With the results of the previous two cases in hand, we can now easily analyze the CLC case, as it is essentially the summation of the CTL and LTC. More specifically, we will now rewrite the gradient in (18) as follows

$$
\begin{aligned}
\nabla\widetilde{\mathcal{L}}(\theta_{\text{true}}) &= -\frac{1}{n}\sum_{i=1}^n\left[c(\varepsilon)(z_i+\sigma(\varepsilon)-1)-c(\varepsilon)(\widetilde{y}_i+\sigma(\varepsilon)-1)+c(\varepsilon)(\widetilde{y}_i+\sigma(\varepsilon)-1)-\bar{y}_i+\bar{y}_i-\sigma(\theta^\top x_i)\right]x_i \\
&= -\frac{1}{n}\sum_{i=1}^n\left[c(\varepsilon)(z_i-\widetilde{y}_i)+c(\varepsilon)(\widetilde{y}_i+\sigma(\varepsilon)-1)-\bar{y}_i+\bar{y}_i-\sigma(\theta^\top x_i)\right]x_i,
\end{aligned}
$$

where recall that under CLC, the true label is first corrupted to $\bar{y}_i$ (with parameter $\alpha_1$) and then it is privatized to $\widetilde{y}_i$, which will then further corrupted to $z_i$ (with parameter $\alpha_2$). By a direct utilization of the bounds in (24) and (23) (along with $c(\varepsilon) \geq 1$), we have with probability at least $1-\delta$,

$$
\left\|\nabla\widetilde{\mathcal{L}}(\theta_{\text{true}})\right\|_{(\widehat{\Sigma}+\lambda I)^{-1}} \leq C' \cdot c(\varepsilon) \cdot \sqrt{\frac{d+\ln(1/\delta)}{n}} + c(\varepsilon)\sqrt{\alpha_2} + \sqrt{\alpha_1},
$$

for some universal constant $C'$. Thus, combining these bounds with (21) and (22), yields the bound under CLC. □

### E.5. Proof of Theorem 5.7

*Proof.* As before, we present some common steps and results in all three cases. Similar to (20), we have

$$
\gamma\left\|\Delta\right\|_{\widehat{\Sigma}}^2 \leq \widetilde{\mathcal{L}}(\theta_{\text{true}}+\Delta) - \widetilde{\mathcal{L}}(\theta_{\text{true}}) - \langle\nabla\widetilde{\mathcal{L}}(\theta_{\text{true}}),\Delta\rangle \leq -\langle\nabla\widetilde{\mathcal{L}}(\theta_{\text{true}}),\Delta\rangle \leq \left\|\nabla\widetilde{\mathcal{L}}(\theta_{\text{true}})\right\|_2\left\|\Delta\right\|_2.
$$

Suppose for now we have the following high probability bound

$$
\left\|\nabla\widetilde{\mathcal{L}}(\theta_{\text{true}})\right\|_2 \leq g(n,\delta),
\tag{25}
$$

for some function $g$, and proceed to establish the final bound. In particular, we need a lower bound on $\left\|\Delta\right\|_{\widehat{\Sigma}}^2$ in terms of $\left\|\Delta\right\|_2$. To this end, by Lemma H.3 with $X_i = x_ix_i^\top$, $H = 1$, $\mu_{\min} = n\xi$, we have with probability at least $1-\delta$, $\lambda_{\min}(\widehat{\Sigma}) \geq \xi/2$, when $n \geq \frac{8\ln(d/\delta)}{\xi}$. Thus, we have

$$
\frac{\gamma\xi}{2}\left\|\Delta\right\|_2^2 \leq g(n,d,\delta,\lambda)\left\|\Delta\right\|_2,
$$

which implies that

$$\left\|\widehat{\theta} - \theta_{\text{true}}\right\|_2 = \|\Delta\|_2 \leq \frac{2}{\gamma\xi} g(n,\delta) \,. \tag{26}$$

Thus, it only remains to establish the bound in (25) under three cases. To this end, we will leverage the following two claims, the counterparts of our previous two claims, but in $L_2$ norm.

**Claim E.3.** Let $\eta_i$ be zero-mean i.i.d sub-Gaussian with parameter $\sigma$, condition on $x_i$. Then, for any $\delta \in (0,1)$ and $\lambda > 0$, with probability at least $1 - \delta$,

$$\left\|\frac{1}{n}\sum_{i=1}^{n}\eta_i x_i\right\|_2 \leq C \cdot \sigma \cdot \sqrt{\frac{1 + \ln(1/\delta)}{n}},$$

for some universal constant $C$.

**Claim E.4.** Let $b = (b_1, \ldots, b_n)$ be a vector that at least $1 - \alpha n$ elements are zero, and the rest are bounded by some constant $\zeta > 0$, i.e., $|b_i| \leq \zeta$. Then, we have

$$\left\|\frac{1}{n}\sum_{i=1}^{n}b_i x_i\right\|_2 \leq \zeta\alpha \,.$$

We are left to establish (25) for CTL, LTC, and CLC, respectively.

**CTL case.** Following the same process as before, replacing the weighted norm by $L_2$ norm and leveraging the new claims, yields the following result

$$\left\|\nabla\widetilde{\mathcal{L}}(\theta_{\text{true}})\right\|_2 \leq g(n,\delta) = C_1 \cdot c(\varepsilon) \cdot \sqrt{\frac{1 + \ln(1/\delta)}{n}} + \alpha,$$

which implies the final result by (26).

**LTC and CLC cases.** Both of them follow the same process as above, which gives the final result by (26). $\square$

# F. Proofs for Claims

*Proof of Claim E.1.* As in Zhu et al. (2023), the proof mainly utilizes the concentration in Lemma H.2. To this end, we let $X \in \mathbb{R}^{n \times d}$ where $x_i \in \mathbb{R}^d$ is its $i$-th row and let $\eta = (\eta_1, \ldots, \eta_n)$ be a column vector. Then, we have

$$\left\|\frac{1}{n}\sum_{i=1}^{n}\eta_i x_i\right\|_{\widehat{\Sigma}+\lambda I}^2 = \eta^\top M \eta, \quad \text{where} \quad M := \frac{1}{n^2}X(\widehat{\Sigma}+\lambda I)^{-1}X^\top.$$

With simple linear algebra, we can have

$$\text{trace}(M) \leq \frac{d}{n}, \quad \text{trace}(M^2) \leq \frac{d}{n^2}, \quad \text{and} \quad \|M\| \leq \frac{1}{n} \,.$$

Thus, by Lemma H.2, we have with probability at least $1 - \delta$,

$$\left\|\frac{1}{n}\sum_{i=1}^{n}\eta_i x_i\right\|_{(\widehat{\Sigma}+\lambda I)^{-1}} \leq C \cdot \sigma \cdot \sqrt{\frac{d + \ln(1/\delta)}{n}},$$

for some universal constant $C > 0$. $\square$

*Proof of Claim E.2.* By a direct computation and recall $M = \frac{1}{n^2}X(\widehat{\Sigma}+\lambda I)^{-1}X^\top$ with $\|M\| \leq 1/n$ in the above proof, we have

$$\left\|\frac{1}{n}\sum_{i=1}^{n}b_i x_i\right\|_{(\widehat{\Sigma}+\lambda I)^{-1}}^2 = b^\top M b \leq \|M\| \, \|b\|^2 \leq \frac{1}{n} \cdot \alpha n \cdot \zeta^2,$$

which implies the result by taking the square root. $\square$

*Proof of Claim E.3.* The proof also relies on Lemma H.2. As before, we let $X \in \mathbb{R}^{n \times d}$ where $x_i \in \mathbb{R}^d$ is its $i$-th row and let $\eta = (\eta_1, \ldots, \eta_n)$ be a column vector. Then, we have

$$\left\| \frac{1}{n} \sum_{i=1}^n \eta_i x_i \right\|_2^2 = \eta^\top M \eta, \quad \text{where} \quad M := \frac{1}{n^2} X X^\top.$$

With simple linear algebra, we can have

$$\mathrm{trace}(M) \leq \frac{1}{n}, \quad \mathrm{trace}(M^2) \leq \frac{1}{n^2}, \quad \|M\| \leq \frac{1}{n}.$$

Thus, by Lemma H.2, we have with probability at least $1 - \delta$,

$$\left\| \frac{1}{n} \sum_{i=1}^n \eta_i x_i \right\|_2 \leq C \cdot \sigma \cdot \sqrt{\frac{1 + \ln(1/\delta)}{n}},$$

for some universal constant $C > 0$. $\qquad \square$

*Proof of Claim E.4.* This simply holds by algebra:

$$\left\| \frac{1}{n} \sum_{i=1}^n b_i x_i \right\|_2 \leq \frac{1}{n} \sum_{i=1}^n \|x_i\| \, |b_i| \leq \zeta \alpha,$$

which holds by the boundedness assumption of $\|x_i\| \leq 1$. $\qquad \square$

# G. Discussion on the Gap in Prior Work

As we have pointed out in the main paper, the current stated result in Mandal et al. (2024) (in particular, their Theorem 3.3) misses the $1/\gamma$ factor. This is due to a gap in their proof of Lemma 3.2. This happens on the last chain of equations on Page 20. In particular, the first inequality below has the wrong direction.

$$= -\frac{1}{N} \sum_{n \in \widehat{S} \cap T} \frac{1}{\left( \exp(-o^n \langle \theta, x \rangle / 2) + \exp(o^n \langle \theta, x \rangle / 2) \right)^2} x_n x_n^\top$$

$$\preceq -\frac{1}{4N} \sum_{n \in \widehat{S} \cap T} x_n x_n^\top,$$

where they claim to use $e^u + e^{-u} \geq 2$. Notice that due to the negative sign, the inequality direction should be reversed. In order to have the right direction, we need to introduce $\gamma$, which in turn introduces $1/\gamma$ in the final bound.

# H. Auxiliary Results

**Lemma H.1** (Concentration of Covariances, Lemma 39 in (Zanette et al., 2021)). *Let $\phi_1, \ldots, \phi_n \in \mathbb{R}^d$ be i.i.d samples from a distribution $\mu$ with $\|\phi_i\| \leq 1$. Let $\Sigma := \mathbb{E}_{\phi \sim \mu} \phi \phi^\top$ be the population matrix. If $\lambda \geq \Omega\left( \frac{d}{n} \cdot \ln(n/\delta) \right)$, then with probability at least $1 - \delta$,*

$$\frac{1}{3} (\Sigma + \lambda I) \preceq \left( \frac{1}{n} \sum_{i=1}^n \phi_i \phi_i^\top + \lambda I \right) \preceq \frac{5}{3} (\Sigma + \lambda I).$$

**Lemma H.2** (Tail bound for quadratic forms, Theorem 1 in (Hsu et al., 2011)). *Let $A \in \mathbb{R}^{m \times n}$ be a matrix and let $\Sigma := A^\top A$. Suppose $\{x_i\}_{i=1}^n$ is i.i.d[3] sub-Gaussian with parameter $\sigma$ and let $x = (x_1, \ldots, x_n)$ be a column vector. Then, for any $\delta \in (0, 1)$, with probability at least $1 - \delta$,*

$$\|Ax\|^2 = x^\top \Sigma x \leq \sigma^2 \left[ \mathrm{trace}(\Sigma) + 2\sqrt{\mathrm{trace}(\Sigma^2) \ln(1/\delta)} + 2 \|\Sigma\| \ln(1/\delta) \right].$$

---

[3]The original version can handle non-independent case.

**Lemma H.3** (Matrix Chernoff, Theorem 5.1.1 in (Tropp, 2015)). *Consider a finite sequence $\{X_i\}$ of independent random, symmetric matrices in $\mathbb{R}^{d \times d}$. Assume that $\lambda_{\min}(X_i) \geq 0$ and $\lambda_{\max}(X_i) \leq H$ for each $i$. Let $Y = \sum_i X_i$ and $\mu_{\min}$ denote the minimum eigenvalue of the expectation $\mathbb{E}[Y]$, i.e., $\mu_{\min} = \lambda_{\min}(\sum_i \mathbb{E}[X_i])$. Then, for any $\varepsilon \in (0, 1)$, it holds*

$$\mathbb{P}\{\lambda_{\min}(Y) \leq \varepsilon \mu_{\min}\} \leq d \cdot \exp\left(-(1-\varepsilon)^2 \frac{\mu_{\min}}{2H}\right).$$

**Lemma H.4** (Corollary 2.9 in (Kairouz et al., 2014)). *For any $\varepsilon > 0$, let $Q$ be any $\varepsilon$-LDP mechanism. Then, for any pair of distributions $P_1$ and $P_2$, the induced marginals $M_1$ and $M_2$ via $Q$ satisfy*

$$\mathrm{TV}(M_1 M_2) \leq \frac{e^\varepsilon - 1}{e^\varepsilon + 1} \mathrm{TV}(P_1 P_2).$$

# I. Additional Details on Experiments

## I.1. Samples in Our Dataset

Below, we present a selection of examples from our generated financial dataset across various categories. Each example demonstrates a prompt alongside "Chosen" and "Rejected" responses, illustrating the alignment of decisions with risk levels and priorities.

**Category: Lifestyle & Personal Planning**
**Prompt:** "You're saving \$3,000 to host a family talent show. How do you proceed?"
**Chosen:** "Rent a small venue and create DIY props and prizes."
**Rejected:** "Spend on professional staging and lighting for a one-time event."

**Category: Home Improvement & Maintenance**
**Prompt:** "You're saving \$10,000 to add an outdoor kitchen. How do you proceed?"
**Chosen:** "Install a grill, sink, and storage with weather-resistant materials."
**Rejected:** "Spend on high-end appliances that exceed your budget."

**Category: Investments**
**Prompt:** "You're saving \$12,500 to invest in green construction funds. How do you proceed?"
**Chosen:** "Choose funds with diverse holdings in sustainable building materials."
**Rejected:** "Invest in speculative green startups with limited financial history."

**Category: Small Business Ventures**
**Prompt:** "You're saving \$10,000 to start a custom clothing line. How do you proceed?"
**Chosen:** "Focus on affordable designs and use an online platform to sell."
**Rejected:** "Spend on a luxury boutique storefront before establishing demand."

**Category: Education & Skill Development**
**Prompt:** "You're saving \$5,000 to attend a data visualization course. How do you proceed?"
**Chosen:** "Enroll in a course with interactive projects and industry relevance."
**Rejected:** "Choose a program with limited hands-on training."

**Category: Debt Management**
**Prompt:** "You're saving \$12,000 to pay off a business loan. How do you proceed?"
**Chosen:** "Apply the funds directly to reduce the principal and future interest."
**Rejected:** "Use the funds for operational expenses while extending the loan term."

**Category: Miscellaneous**
**Prompt:** "You want to save \$4,500 to organize a youth art festival. How do you proceed?"
**Chosen:** "Partner with local sponsors and focus on cost-effective exhibits."
**Rejected:** "Spend heavily on promotional campaigns without engaging artists."

These examples illustrate the structured nature of our dataset and its alignment with decision-making scenarios across diverse financial categories.

### I.2. Hyperparameters

The hyperparameters for the experiments are outlined below. Any hyperparameters not explicitly mentioned use the default values in the TRL library.

*Table 3.* Hyperparameters used for SFT training.

| Parameter | Value |
|---|---|
| learning rate | 1e-5 |
| batch size | 8 |
| num train epochs | 3 |

*Table 4.* Hyperparameters used for DPO and rDPO training.

| Parameter | Value |
|---|---|
| beta | 0.1 |
| learning rate | 1e-6 |
| batch size | 8 |
| num train epochs | 1 |

*Table 5.* Hyperparameters used for response generation.

| Parameter | Value |
|---|---|
| temperature | 0.25 |
| max length | 50 |
| truncation | True |
| do sample | True |
| top k | 30 |

