# OpenReview forum: "A Unified Theoretical Analysis of Private and Robust Offline Alignment: from RLHF to DPO"
_ICML.cc/2025/Conference — ICML 2025 spotlightposter_

### Official Review · Reviewer_hRoJ · 2025-03-13

**Overall Recommendation:** 3

**Summary:**

This submission studies the interplay between privacy and robustness in both RLHF and DPO, two main alignment methods for language models. It shows that, when considering a linear reward model for RLHF and a log-linear policy class for DPO, the problem of offline alignment reduces to parameter estimation for logistic regression under private and corrupted labels. Using this reduction, one can derive suboptimality upperbound for the policy learned for both RLHF and DPO, based on parameter estimation error for logistic regression under private and corrupted labels.

Summary after rebuttal: I asked a question from the authors after their rebuttal about their considered corruption model in the LTC case. However, the authors did not reply to this question. Yet, I am going to keep my score for the other results in the paper.

**Claims And Evidence:**

The authors have provided proofs for the theoretical claims made in the submission.

**Essential References Not Discussed:**

It seems that most of the related works are mentioned and discussed in the submission.

**Experimental Designs Or Analyses:**

I have the following comments/questions about the experimental results.

1. I think it would be better if the authors could manage to include the experimental results (Tables 1 and 2 in the appendix) in the main body of the paper.

2. Also, why are there no results included on RLHF under privacy/corruption/both? For completeness, both DPO and RLHF should be evaluated under four scenarios of privatization (RR)/corruption/LTC/TCL.

**Methods And Evaluation Criteria:**

Currently, the experiments reported in the submission are very limited in terms of the amount of experimental results, the datasets and the used alignment algorithms. The only reported results are in Table 1 and 2, which are just for DPO (and rDPO). No results are reported for RLHF under corrupted/privatized labels.

**Other Comments Or Suggestions:**

see above

**Other Strengths And Weaknesses:**

I have mentioned my comments/questions above. After receiving the authors' answers, if needed, I will consider changing my evaluation.

**Questions For Authors:**

1. Is there any intuitive reason why alignment under LTC is more challenging than CTL? For example, in the first row of Table. 2, $\epsilon=1$ and $\alpha=0.1$. Therefore, in LTC, each label is flipped by randomization (RR) with probability 0.27 followed by another flip due to corruption with probability $\alpha=0.1$. Therefore, roughly, the probability of each label to get flipped (compared to that in the clean preference dataset) is 0.027. With the same reasoning, the probability of each label getting flipped under TCL is also roughly 0.027. So why does the order of the two label flippings (RR and corruption) matter? From another point of view: the constant $c(\epsilon)$ appearing in the upperbound of LTC is not a large constant (for example for $\epsilon=1$, it is 2.16). However, from the results in Table. 2, we see that LTC is clearly worse than CTL. Do the authors have an intuitive idea why this is the case?

2. The main assumptions made in this submission are 1. a linear reward model for RLHF 2. a log-linear policy model for DPO 3. The label distribution in the preference dataset following a LR model. How much limitative these assumptions are? Can we extend the obtained theoretical results to other RLHF/DPO models with more relaxed assumptions on the data as well as reward/policy models?

**Relation To Broader Scientific Literature:**

In the context of alignment of language models, this work allows for establishing a theoretical separation result between LTC and CTL, which is important to consider when performing offline alignment of language models.

**Theoretical Claims:**

I did not go through the proofs in details. However, I had a quick look at them to get an idea of their sketch.

---

> ### Author Rebuttal · Authors · 2025-03-30
>
> Thanks for your time and comments. We will recap your valuable comments and present our detailed response. We hope our answers will resolve your concern.
>
> **1. About experiments.** Thank you for your valuable suggestion. In the next revision, we will aim to include our experiments within the main body. Since our primary contributions are theoretical, we initially chose to present experiments using DPO as a proof of concept. However, we believe similar results can be obtained with RLHF, aligning with the findings in [R1], which considers corruption alone. We plan to incorporate additional results on RLHF in our future revision.
>
> **2. Intuition about the separation between LTC and CTL.** Your observation regarding the random-flipping corruption model is absolutely correct. That is, under this oblivious corruption model, the final corruption rate is the same between LTC and CTL. However, it becomes different when we move from this corruption model. Think about the following situation. The clean data is all 0, and the corruption model here is to simply set the data to be 1 (with probability $\alpha$). That is, with probability $\alpha$, the data will be set to be 1, otherwise, it remains unchanged. This is exactly the corruption model considered in our experiment, which is different from the random flipping corruption model. As mentioned in Line 737, one can see that LTC will lead to more 1s in the end compared to CTL (as the randomized response of LDP may flip back 1 to 0).
>
> **3. Generalization beyond current assumptions.** Thank you for the insightful question. Although our main results (e.g., the separation between LTC and CTL) are established under our current assumptions, we believe these findings will extend more generally—beyond linear models—to broader reward functions, policy classes, and general preference models (not limited to the BT-model). However, transitioning from linear to more general settings will likely require new analytical techniques. This challenge arises because, even in settings without privacy or corruption, employing an analysis similar to ours yields suboptimal rates [R2]. Exploring these generalizations represents an exciting avenue for future research. Nevertheless, we believe that our current results under the linear model will serve as important benchmarks for these advancements.
>
> ---
>
> [R1] Chowdhury, S. R., Kini, A., and Natarajan, N. Provably robust DPO: Aligning language models with noisy feedback. arXiv preprint arXiv:2403.00409, 2024
>
> [R2] Zhu, B., Jordan, M., and Jiao, J. Principled reinforcement learning with human feedback from pairwise or K-wise comparisons. In International Conference on Machine Learning, pp. 43037–43067. PMLR, 2023.

---

### Official Review · Reviewer_2Ymr · 2025-03-13

**Overall Recommendation:** 4

**Summary:**

The paper considers alignment problems such as RLHF or DPO, in a robust private setting. In this setting, we are given a preference dataset where each example contains an input text $s$, two actions $a_0, a_1$, and a label $y \in \{0, 1\}$ denoting which action is preferred in the example. Commonly, we assume the label $i$ is sampled with probability proportional to $exp(r(s, a_i))$, where $r$ is a reward function on the text $s$ and action $a$ In RLHF we train an intermediate reward model on the dataset which is then used to optimize the policy, in DPO the policy is directly optimized on the dataset. In the robust, private setting there are two sources of error on the labels (1) corruption, where some fraction $\alpha$ of the labels are corrupted by an adversary after inspecting the dataset, and (2) local DP, where we apply randomized response to change the label of each example (with probability $1 / (1 + e^\epsilon)$ for some $\epsilon$) to provide privacy to user. One can consider the LTC or CTL settings, where local DP is applied and then corruption occurs, or vice-versa. The authors give an analysis for both RLHF and DPO, in the LTC, CTL, and "CLC" setting (where corruption occurs both before and after local DP).

For RLHF the analysis assumes that the reward function is the dot product of some ground (bounded norm) ground truth $\theta^*$ and some known function $\phi(s, a)$, which was done by most of the prior work. They consider two algorithms: One that chooses a policy maximizing the expectation over the policy of this dot product for a current parameter estimate of $\theta^*$, and one that choosing a policy that maximizes the minimum this expectation can increase relative to a reference policy even if the parameter estimate is perturbed by a bounded amount. Under the linear model assumption, the authors show the labels correspond to a logistic regression model and bound the suboptimality of their algorithms. For DPO, they assume the optimal policy falls a log-linear model and prove similarly that the labels in DPO follow a logistic regression model and bound the suboptimality of a policy that is a function of an estimate of the ground truth in the logistic regression model.

Using these results, the authors design algorithms for getting the parameter estimate of $\theta^*$, by shifting and scaling the received labels based on the privacy of randomized response. They give bounds for this parameter estimate's error, with and without corruption, and with and without a uniform coverage assumption (the covariance of the features has lower bounded eigenvalues). Combining the parameter estimation bounds with the suboptimality bounds for the algorithm using the parameter estimate, they derive suboptimality bounds for RLHF and DPO in the LTC and CTL settings, with or without uniform converage. The bounds consist of (1) a bias term depending on the corruption parameter $\alpha$, which is larger for LTC than for CTL, demonstrating that LTC is fundamentally harder) and (2) terms decaying to 0 as $1/\sqrt{n}$ in the dataset size $n$, matching the noiseless rate.

## update after rebuttal
I remain in support of accepting the paper

**Claims And Evidence:**

Yes

**Essential References Not Discussed:**

N/A

**Experimental Designs Or Analyses:**

Yes, no major issues.

**Methods And Evaluation Criteria:**

Yes

**Other Comments Or Suggestions:**

-Line 24, column 2: Missing period
-Line 265, column 1: "singple" -> single

**Other Strengths And Weaknesses:**

Other Strengths:
* The technical results are reasonably difficult to arrive at.
* Presentation allows for one who is not super familiar with RLHF/DPO or with local DP/randomized response to understand the problem setup and algorithms reasonably well.

Other Weaknesses:
* A minor shared weakness with past work, that the analysis assumes a linear reward function. The authors do cite this as room for improvement in future works, and attempt to mitigate this with empirical studies where the linear reward function assumption may not hold.
* Minor presentation weakness: The introduction suggests a separation, which implies a lower bound for one setting and upper bound for the other setting. However, the results are upper bounds only. It is unlikely but theoretically possible there is no separation with a different algorithm, this should be clarified when discussing the contributions of the paper.

**Questions For Authors:**

No questions that would substantially affect my evaluation.

**Relation To Broader Scientific Literature:**

Matches the dependence on $n$ of past work in the noiseless setting. Improves on Mandal et al. in terms of dependence on corruption parameter $\alpha$ for RLHF under a weaker corruption model, and matches the offline rate. For DPO, improves dependence on $\alpha$ and removes dimension dependence for bias term. First result on private and robust DPO, prior work studied private-only or robust-only. Improves Chowdury et al. result for robust DPO which has weaker dependence of $1/n^{1/4}$ on the dataset size. Overall, seems to match the rate in more restrictive settings (i.e. non-private or non-robust) or improve the rates of past works.

**Theoretical Claims:**

I did not check the proofs in detail.

---

> ### Author Rebuttal · Authors · 2025-03-30
>
> Thanks for your positive feedback. We will fix the typos in the next version. We now recap your comment and present our detailed response.
>
> **Minor presentation weakness about LTC and CTL.** Thanks for your sharp question. You are absolutely right and we will make this point more clear in the paper. We have briefly talked about the lower bound on the separation between LTC and CTL in the conclusion section and Appendix C. Let's recap them based on the following two scenarios:
>
> - **With uniform coverage condition.** As mentioned in Appendix C, we believe that the current additional $c(\epsilon)$ faction in LTC is tight.
>
> - **Without uniform coverage condition.** As mentioned in the conclusion section, in this case, the current additional factor in LTC may not be tight. However, we believe that there still exists a separation between LTC and CTL. The intuition here is that adversary corruption after randomized response for privacy can always lead to a larger amount of flipping of the preference labels, as the adversary can focus on the attacks on the clean data by first observing the output of LDP.

---

### Official Review · Reviewer_FQRv · 2025-03-14

**Overall Recommendation:** 4

**Summary:**

The authors provide a theoretical framework to analyze the suboptimality gap of the learned policy in offline alignment, in the presence of privatized and corrupted labels. Specifically, they reduce two main paradigms, the Reinforcement Learning from Human Feedback (RLHF) and Direct Preference Optimization (DPO), to the parameter estimation problem of logistic regression under some linear model assumptions. The suboptimality gap is then upper bounded by the estimation error of the logistic regression parameters. They demonstrate that, with a “shifting and scaling” loss function, the estimation error can be upper-bounded when labels are privatized (i.e., random response mechanism) and corrupted (i.e., an adversary modifies partial labels arbitrarily), quantifying the robustness of the alignment training to noisy labels.

**Claims And Evidence:**

The claims made in the paper (“a unified analysis of the interplay between
privacy and robustness in both RLHF and DPO”) are well matched with their theoretical results.

**Essential References Not Discussed:**

N/A

**Experimental Designs Or Analyses:**

There are no experiments included in the main text. One experiment is included in the Appendix, showing that training with a robust loss function to noisy labels can outperform the regular cross entropy loss, which demonstrates the benefits of robust alignment training. This aligns with the robustness shown in the paper.

**Methods And Evaluation Criteria:**

Reducing the alignment problem to logistic regression and upper bounding the estimation error is a promising and sensible approach.

**Other Comments Or Suggestions:**

N/A

**Other Strengths And Weaknesses:**

Other strength: as the authors point out, the results of the estimation error bounds of private/robust logistic regression may be of independent interest.

**Questions For Authors:**

N/A

**Relation To Broader Scientific Literature:**

N/A

**Theoretical Claims:**

I do not check the correctness of the proof, but it reads technically sound.

---

> ### Author Rebuttal · Authors · 2025-03-30
>
> Thank you for the positive evaluation of our paper. We are delighted to hear that you find our methods promising and well-founded.

---

### Official Review · Reviewer_xT8c · 2025-03-22

**Overall Recommendation:** 4

**Summary:**

The paper develops a unified theoretical framework analyzing the impact of label corruption and privacy on two primary offline alignment methods: RLHF and DPO. The authors focus on the interplay between LDP and adversarial label corruption, formalizing three noise models: CTL (corruption then LDP), LTC (LDP then corruption), and CLC (corruption before and after LDP). A key conceptual contribution is a reduction from the offline alignment problem to logistic regression under these noise models. Major findings include:

- LTC is provably more challenging than CTL, even under linear models.
- The analysis yields new state-of-the-art theoretical guarantees for offline alignment under privacy-only or corruption-only regimes.
- Novel estimation bounds are derived for logistic regression under the joint noise setting, leading to suboptimality bounds for RLHF and DPO algorithms.
- The authors also provide practical algorithmic implications, including a new estimator and empirical verification of the separation between LTC and CTL on GPT2.

**Claims And Evidence:**

The main claims of the paper—such as the separation between CTL and LTC, and the improved suboptimality bounds under joint privacy and corruption—are supported by a well-structured reduction to logistic regression under noisy labels. While the claims appear to be justified by clear derivations and formal proofs, I have not carefully reviewed the correctness of the theoretical arguments.

**Essential References Not Discussed:**

NA

**Experimental Designs Or Analyses:**

While the primary focus is theoretical, the authors include empirical experiments on GPT2-large that demonstrate the practical difference between CTL and LTC. These experiments are not extensively described in the main paper, but are useful as sanity checks. Further empirical validation across more complex or realistic settings would strengthen the practical case, but is not strictly necessary for this theory-focused submission.

**Methods And Evaluation Criteria:**

The methodological choice to reduce the suboptimality of RLHF and DPO to parameter estimation in logistic regression is well-motivated and principled, enabling a unified treatment of privacy and corruption. The authors assess performance using standard metrics, including suboptimality gap and parameter estimation error in both weighted and Euclidean norms, all under clearly stated assumptions such as linear reward models and bounded feature maps. These choices appear appropriate given the theoretical focus of the work. That said, due to my limited familiarity with several of the domains involved, I am not in a position to rigorously evaluate the methodology in its entirety.

**Other Comments Or Suggestions:**

NA

**Other Strengths And Weaknesses:**

NA

**Questions For Authors:**

NA

**Relation To Broader Scientific Literature:**

From the perspective of the LDP literature, this paper contributes a novel extension of LDP techniques to preference-based alignment, which remains relatively underexplored in the privacy literature. The authors extend ideas from private logistic regression and randomized response mechanisms, as seen in works such as Chowdhury et al. (2023), to a more complex setting involving both privacy and adversarial corruption.

While I do not have deep familiarity with the broader literature on the other areas covered in the paper, the authors position their work as unifying and extending previous theoretical treatments of RLHF and DPO under separate noise models. Their comparison between CTL and LTC, and the development of a unified reduction to private and corrupted logistic regression, appear to be novel contributions that bridge multiple subfields.

**Theoretical Claims:**

The derivations appear technically sound from a high level, and the proofs are well-organized. However, I am not equipped to verify their correctness in detail. I particularly appreciate the clean and interpretable separation between CTL and LTC in the error bounds, which draws an interesting parallel to similar phenomena observed in private robust mean estimation.

---

> ### Author Rebuttal · Authors · 2025-03-30
>
> Thank you for your time and positive evaluation of our paper. We're glad to hear that you found our methods well-motivated, our derivations clear, and our experimental results useful.

---

### Decision · Program_Chairs · 2025-05-01

**Decision:**

Accept (spotlight poster)

**Comment:**

This paper presents a unified and rigorous theory to understand the impact of label corruption and local differential privacy on offline alignment methods including RLHF and DPO. All reviewers recommend acceptance due to the paper’s novel reduction to logistic regression under joint noise, insightful separation between CTL and LTC settings, and new bounds that advance the literature. One limitation is that the current result heavily relies on the linear-model assumption.  Overall, the theoretical contribution still makes it a strong submission that deserves acceptance.